# The stabilized supralinear network accounts for the contrast dependence of visual cortical gamma oscillations

Caleb J. Holt[1], Kenneth D. Miller[2], Yashar Ahmadian[3]*

**1** Department of Physics, Institute of Neuroscience, University of Oregon, Eugene, Oregon, United States of America, **2** Deptartment of Neuroscience, Center for Theoretical Neuroscience, Swartz Program in Theoretical Neuroscience, Kavli Institute for Brain Science, College of Physicians and Surgeons, and Morton B. Zuckerman Mind Brain Behavior Institute, Columbia University, New York, New York, United States of America, **3** Department of Engineering, Computational and Biological Learning Lab, University of Cambridge, Cambridge, United Kingdom

\* ya311@cam.ac.uk

**Data Availability Statement:** A code package sufficient for producing the results of this paper (including a script for the samplings of models and calculations of their LFP power-spectra) can be

## Abstract

When stimulated, neural populations in the visual cortex exhibit fast rhythmic activity with frequencies in the gamma band (30-80 Hz). The gamma rhythm manifests as a broad resonance peak in the power-spectrum of recorded local field potentials, which exhibits various stimulus dependencies. In particular, in macaque primary visual cortex (V1), the gamma peak frequency increases with increasing stimulus contrast. Moreover, this contrast dependence is local: when contrast varies smoothly over visual space, the gamma peak frequency in each cortical column is controlled by the local contrast in that column's receptive field. No parsimonious mechanistic explanation for these contrast dependencies of V1 gamma oscillations has been proposed. The stabilized supralinear network (SSN) is a mechanistic model of cortical circuits that has accounted for a range of visual cortical response nonlinearities and contextual modulations, as well as their contrast dependence. Here, we begin by showing that a reduced SSN model without retinotopy robustly captures the contrast dependence of gamma peak frequency, and provides a mechanistic explanation for this effect based on the observed non-saturating and supralinear input-output function of V1 neurons. Given this result, the local dependence on contrast can trivially be captured in a retinotopic SSN which however lacks horizontal synaptic connections between its cortical columns. However, long-range horizontal connections in V1 are in fact strong, and underlie contextual modulation effects such as surround suppression. We thus explored whether a retinotopically organized SSN model of V1 with strong excitatory horizontal connections can exhibit both surround suppression and the local contrast dependence of gamma peak frequency. We found that retinotopic SSNs can account for both effects, but only when the horizontal excitatory projections are composed of two components with different patterns of spatial falloff with distance: a short-range component that only targets the source column, combined with a long-range component that targets columns neighboring the source column. We thus make a specific qualitative prediction for the spatial structure of horizontal connections in macaque V1, consistent with the columnar structure of cortex.

found at https://gitlab.com/ahmadianlab/ssn_gamma.

**Funding:** YA is supported by UK Research and Innovation, Biotechnology and Biological Sciences Research Council (www.ukri.org/councils/bbsrc) research grant BB/X013235/1. KDM is supported by National Science Foundation (www.nsf.gov) grant DBI-1707398, National Institutes of Health (www.nih.gov) grants U01NS108683, R01EY029999, and U19NS107613, Simons Foundation (www.simonsfoundation.org) award 543017, and the Gatsby Charitable Foundation (www.gatsby.org.uk). The funders had no role in study design, data collection and analysis, decision to publish, or preparation of the manuscript.

**Competing interests:** The authors have declared that no competing interests exist.

## Author summary

When large populations of brain's neurons fire in synchrony, the resulting electrical signals, which are measurable even at the scalp, exhibit oscillatory behaviour. Gamma rhythms are a fast sub-type of such oscillations generated by neural populations in visual processing areas of the cerebral cortex. The characteristics of gamma oscillations depend on the visual scene observed by the animal. For example, visual stimuli with stronger contrast evoke higher frequency oscillations. Moreover, when contrast varies over the visual scene, signals measured at different cortical locations oscillate at frequencies determined by the local contrasts in the corresponding visual scene locations. The circuit mechanisms underlying these phenomena are largely unknown. Here, we show how a model of cortical circuits explains the contrast dependence of gamma frequency as arising from the empirical observation that a given percent increase in the input to a cortical neuron results in a higher percent increase in its output. Furthermore, the model imputes the local contrast dependence of gamma frequency to a certain spatial pattern of connectivity between cortical neurons. Relating properties of an easily measurable brain signal to features of brain circuits can link anomalies in the two, which may potentially be exploited in diagnosis of brain disorders.

## Introduction

When presented with a stimulus, populations of neurons within visual cortices exhibit elevated rhythmic activity with frequencies in the so-called gamma band (30–80 Hz) [1, 2]. These gamma oscillations can be observed in local field potential (LFP) or electroencephalogram (EEG) recordings and, when present, manifest as peaks in the LFP/EEG power-spectra. It has been proposed that gamma oscillations perform key functions in neural processing such as feature binding [3], dynamic communication or routing between cortical areas [4–7], or as a timing or "clock" mechanism that can enable coding by spike timing [8–12]. These proposals remain controversial [1, 13].

While the computational role of gamma rhythms is not well understood, much is known about their phenomenology. For example, defining characteristics of gamma oscillations, such as the width and height of the spectral gamma peak, as well as its location on the frequency axis (peak frequency), exhibit systematic dependencies on various stimulus parameters [1, 2, 14, 15]. In particular, in the primary visual cortex (V1) of macaque monkeys, the power-spectrum gamma peak moves to higher frequencies as the contrast of a large and uniform grating stimulus is increased [1, 2]. This establishes a monotonic relationship between gamma peak frequency and the grating contrast. We will refer to this contrast-frequency relationship, obtained using a grating stimulus with uniform contrast, as the "contrast dependence" of gamma peak frequency.

Moreover, when animals are presented with a stimulus with non-uniform contrast that varies over the visual field (and hence over nearby cortical columns in V1), it is the local stimulus contrast that determines the peak frequency of gamma oscillations at a cortical location [1]. Specifically, [1] used a Gabor stimulus (which has smoothly decaying contrast with increasing distance from the stimulus center), and found that the gamma peak frequency of different V1 recording sites match the predictions resulting from the frequency-contrast relationship obtained from the uniform grating experiment, but using the local Gabor contrast in that site's

receptive field. We refer to this second effect as the "local contrast dependence" of gamma peak frequency.

It is well-known that networks of excitatory and inhibitory neurons with biologically realistic neural and synaptic time-constants can exhibit oscillations with frequency in the gamma band (*e.g.*, [16, 17]; see [18] for a review). However, no mechanistic circuit model of visual cortex has been proposed which can robustly and comprehensively account for the contrast dependence of gamma oscillations. [2] did propose a rate model that accounts for the increase of gamma peak frequency with increasing global contrast. Their treatment only modeled the interactions between a single excitatory and a single inhibitory population, which is sufficient for spatially uniform stimuli, but cannot explain the local contrast dependence of the gamma peak frequency. Moreover, even in the case of a uniform-contrast stimulus, this model could only produce very weak contrast-dependence of peak frequency, and further required a contrast-dependent scaling of the intrinsic time-constant of excitatory neurons. Here, we develop a parsimonious and self-contained mechanistic model (with fixed neural and network parameters) which accounts for the global as well as local contrast dependence of the gamma peak.

It is not clear how the local contrast dependence of gamma oscillations can be reconciled with key features of cortical circuits. This locality would trivially emerge if cortical columns were non- or weakly interacting; in that case each column's oscillation properties would be determined by its feedforward input (controlled by the local contrast). However, nearby cortical columns do interact strongly via the prominent horizontal connections connecting them [19]. These interactions manifest, *e.g.*, in contextual modulations of V1 responses, such as in surround suppression [20], which are thought to be partly mediated by horizontal connections [21].

Surround suppression is the phenomena wherein stimuli outside the classical receptive field (RF) of V1 neurons, which by themselves cannot drive the cell to respond, nevertheless modulate the cells' response, typically by suppressing it. Surround suppression results in a non-monotonic "size tuning curve", which is obtained by measuring a cell's response to circular gratings of varying sizes centered on that cell's RF: the response first increases with increasing stimulus size, but then decreases as the grating increasingly covers regions surrounding the RF. Here we test whether a model of V1, featuring biologically plausible horizontal connections, can capture both surround suppression and the local contrast dependence of gamma oscillations.

A parsimonious, biologically plausible model of cortical circuitry which has successfully accounted for a range of cortical contextual modulations and their contrast dependence is the stabilized supralinear network (SSN) [22, 23]. In particular, the SSN robustly captures the contrast dependencies of surround suppression, *e.g.*, that size tuning curves peak at smaller stimulus sizes with increasing stimulus contrast [22]. Being a recurrently connected firing rate model with excitatory and inhibitory neurons, we expect the SSN to be able to exhibit oscillations similar to gamma rhythms. However, to capture fast dynamical phenomena, and in particular the gamma band resonance frequency, it is key to properly account for fast synaptic filtering as provided by the fast ionotropic receptors, AMPA and $GABA_A$ [17, 24, 25]. At the same time, it is useful to include the slower NMDA conductances, to help stabilize the network dynamics given strong overall recurrent excitation. We thus started by extending the SSN model to properly account for input currents through different synaptic receptor types, with different filtering timescales.

The synchrony and coherence characteristics of gamma oscillations depend considerably on the stimulus condition. In some conditions, such as for large high-contrast gratings, these oscillations are very pronounced and result in rather sharp peaks in the power spectrum of LFP or multi-unit activity [2, 15]. However, the gamma phase is only auto-coherent over

relatively short intervals (not lasting more than a few periods of oscillation) and their timing and duration vary stochastically [13, 26]. These characteristics are consistent with selective amplification by the network of gamma frequencies in the input noise, rather than a noise-free periodic oscillation, and gamma oscillations have been modelled as such [26–30]. We therefore used a noise-driven SSN with multiple synaptic receptor types to model gamma oscillations.

We start the Results section by developing an extension of the SSN that models the dynamics of input currents through different synaptic receptor types, with different timescales. We then study a reduced noise-driven SSN composed of two units representing excitatory (*E*) and inhibitory (*I*) sub-populations. We show that, for a wide range of biological parameters, this reduced SSN model generates gamma oscillations with peak frequency that robustly increases with increasing external drive to the network. We show that this robust contrast dependence is a consequence of a key feature of the SSN: the supralinear input-output (I/O) function of its neurons (which is known to fit well the non-saturating and expansive relationship between the firing rate and membrane voltage of V1 neurons [31, 32]). We next investigate the gamma peak's local contrast dependence using an expanded retinotopically organized SSN model of V1, with *E* and *I* units in different cortical columns. We show that this network is capable of reproducing the local contrast dependence of gamma peak frequency while exhibiting realistic surround suppression. However, as we show, this is only possible when the spatial fall-off of excitatory connection strengths has two distinct components: a sharp immediate fall across a cortical column's width, and a slower fall off that can range over several columns. This "local plus long-range" spatial structure of horizontal connections, which we will more shortly refer to as "columnar structure", balances the trade-off between capturing local contrast dependence (requiring short-range or weak horizontal connections) and surround suppression (requiring the opposite). We show that achieving this balance does not require fine-tuning of parameters and is robust to considerable parameter variations. We end by providing a mathematical explanation of the mechanism underlying local contrast dependence reconciled with strong surround suppression in this model, based on the structure of its normal oscillatory modes. Finally, in the Discussion, we conclude by discussing the implications of our findings for the structure of cortical horizontal connections and the shape of neural input/output nonlinearities.

## Results

### Noise-driven SSN with multiple synaptic currents

As motivated in the Introduction, and with the aim of modeling gamma oscillations, we started by extending the SSN model to properly account for synaptic currents through different receptor types with different kinetics. In its original form, the SSN's activity dynamics are governed by standard firing rate equations [33], in which each neuron is described by a single dynamical variable: either its output firing rate [22, 23] or its total input current [34]. In the extended model, by contrast, each neuron will have more than one dynamical variable, corresponding to its input currents through different synaptic receptor types. Concretely, we will include the three main ionotropic synaptic receptors in the model: AMPA and NMDA receptors which mediate excitatory inputs, and GABA$_A$ (henceforth abbreviated to GABA) receptors which mediate the inhibitory input. For a network of *N* neurons, we will arrange these input currents to different neurons into three *N*-dimensional vectors, $\mathbf{h}^\alpha$, where $\alpha \in \{$AMPA, NMDA, GABA$\}$ denotes the receptor type. To model the kinetics of different receptors, we will ignore the very fast rise-times of all receptor types (as the corresponding timescales are much faster than the characteristic timescales of gamma oscillations), and only account for the receptor decay-times, which we denote by $\tau^\alpha$. With this assumption, the dynamics of $\mathbf{h}_t^\alpha$ are

governed by (see Methods for a derivation)

$$\tau_\alpha \frac{d\mathbf{h}_t^\alpha}{dt} + \mathbf{h}_t^\alpha = W^\alpha \mathbf{r}_t + \mathbf{I}_t^\alpha \tag{1}$$

where $\mathbf{r}_t$ is the vector of firing rates, $W^\alpha \mathbf{r}_t$ and $\mathbf{I}_t^\alpha$ denote the recurrent and external inputs to the network mediated by receptor $\alpha$, respectively, and $W^\alpha$ are $N \times N$ matrices denoting the contributions of different receptor-types to recurrent connectivity weightsl; the total recurrent connectivity weight matrix is thus given by $W \equiv \sum_\alpha W^\alpha$. As in the cortex, the external input to the network is excitatory, and for simplicity we further assume that it only enters through AMPA receptors (*i.e.* $\mathbf{I}_t^\alpha$ is nonzero only for $\alpha$ = AMPA, and we will thus drop this superscript and denote this input by $\mathbf{I}_t$); including an NMDA components will not affect our results, as NMDA is slow relative to gamma band timescales. To close the system for the dynamical variables $\mathbf{h}_t^\alpha$, we have to relate the output rate of a neuron to its total input current. The fast synaptic filtering provided by AMPA and GABA allows for a static (or instantaneous) approximation to the input-output (I/O) transfer function of neurons [35, 36] (see Methods for further justification of this approximation):

$$\mathbf{r}_t = F(\mathbf{h}_t^{\text{total}}) = F\left(\sum_\beta \mathbf{h}_t^\beta\right), \tag{2}$$

where the I/O function $F(\cdot)$ acts element-wise on its vector argument. As in the original SSN, we take this I/O transfer function to be a supralinear rectified power-law, which is the essential ingredient of the SSN (see Fig 1A inset): $F(v) \equiv k[v]_+^n$, where $k$ is a positive constant, $n > 1$ (corresponding to supralinearity), and $[x]_+ \equiv \max(0, x)$ denotes rectification. While the I/O function of biological neurons saturates at high firing rates (*e.g.*, due to refractoriness), throughout the natural dynamic range of cortical neurons firing rates stay relatively low. In fact, in V1 neurons the relationship between the firing rate and the mean membrane potential (an approximate surrogate for the neuron's net input) shows no saturation throughout the entire range of firing rates driven by visual stimuli, and is well approximated by a supralinear rectified power-law [31, 32].

## Two-population model

We start by studying a reduced two-population model of V1 consisting of two units (or representative mean-field neurons): one excitatory and one inhibitory unit, respectively representing the excitatory and inhibitory neural sub-populations in the retinotopically relevant region of V1. This reduced model is appropriate for studying conditions in which the spatial profile of the activity is irrelevant, *e.g.*, for a full-field grating stimulus where we can assume the relevant V1 network is uniformly activated by the stimulus. Both units receive external inputs, and make reciprocal synaptic connections with each other as well as themselves (Fig 1A).

As pointed out in the Introduction, empirical evidence is most consistent with visual cortical gamma oscillations resulting from noise-driven fluctuations [13, 26, 27, 29]. To model such noise-driven oscillations using the SSN, as in [34], we assumed the external input consists of two terms $\mathbf{I}_t = \mathbf{I}_{DC} + \boldsymbol{\eta}_t$, where $\mathbf{I}_{DC}$ represents the feedforward stimulus drive to the network (by a steady time-independent stimulus) and scales with the contrast of the visual stimulus, and $\boldsymbol{\eta}_t$ represents the stochastic noise input to the network. This input noise could be attributed to several sources, including sources that are (biologically) external or internal to V1. External noise can originate upstream in the lateral geniculate nucleus (LGN) of thalamus, or in feedback from higher areas. Internally generated noise results from the network's own irregular spiking (not explicitly modeled) which survives mean-field averaging as a finite-size effect

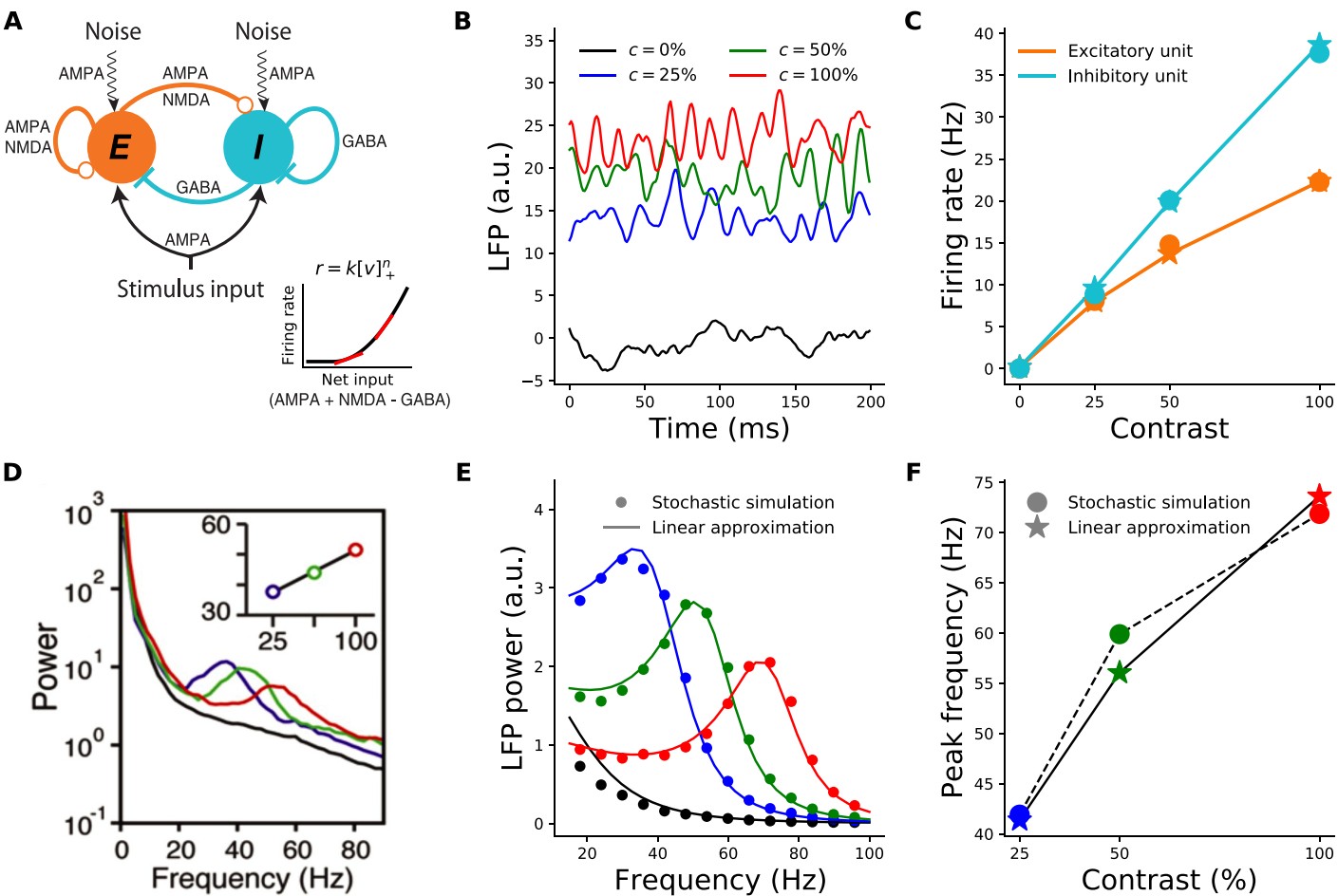

**Fig 1. Contrast dependence of the gamma peak frequency in the 2-population model. A:** Schematic of the 2-population Stabilized Supralinear Network (SSN). Excitatory (*E*) connections end in a circle; inhibitory (*I*) connections end in a line. Each unit represents a sub-population of V1 neurons of the corresponding *E*/*I* type. Both receive inputs from the stimulus, as well as noise input. **Inset:** The rectified power-law Input/Output transfer function of SSN units (black). Red lines indicate the slope of the I/O function at particular locations. **B:** Local field potential (LFP) traces, modeled as total net input to the *E* unit, from the stochastic model simulations under four different stimulus contrasts (*c*): 0% (black) equivalent to no stimulus or spontaneous activity, 25% (blue), 50% (green), 100% (red). The same color scheme for stimulus contrast is used throughout the paper. (Note that we take stimulus strength (input firing rate) to be proportional to contrast, although in reality it is monotonic but sublinear in contrast, [38]). **C:** Mean firing rates of the excitatory (orange) and inhibitory (cyan) units as a function of contrast, from the stochastic simulations (dots) and the noise-free approximation of the fixed point, Eq 3 (stars). Note that the dots and stars closely overlap. **D:** Reproduction of figure 1I from [1] (with permission) showing the average of experimentally measured LFP power-spectra in Macaque V1. The inset shows the dependence of gamma peak frequency on the contrast of the grating stimulus covering the recording site's receptive field. **E:** LFP power-spectra for *c* = 0%, 25%, 50%, 100% (black, blue, green, and red curves, respectively) calculated from the noise-driven stochastic SSN simulations (dots), or using the linearized approximation (solid lines). **F:** Gamma peak frequency as a function of contrast, obtained from power-spectra calculated using stochastic simulations (dots and dashed line) or the linearized approximation (stars and solid line).

(given the finite size of the implicitly-modeled spiking neuron sub-populations underlying the SSN's units). For parsimony, we assumed noise statistics are independent of stimuli, and thus of contrast. (Internally generated spiking noise is expected to have power that grows with the emerging firing rates in the network, as in a Poisson process. Since we are not interested in modeling changes in the gamma power —as opposed to peak frequency— with increasing contrast, we ignored this scaling, as it would not qualitatively affect the contrast dependence of peak frequency). More specifically, we assumed that noise inputs to different neurons are independent, and each component is temporally correlated pink noise with a correlation time on the order of a few milliseconds (our main results are robust to changes in this parameter, as well as to the introduction of input noise correlation across neurons).

For the first results shown in Fig 1, we directly simulated the stochastic Eq 1. Fig 1C (dots) shows the average firing rates found in these simulations, and their contrast-dependence. The LFP signal is thought to result primarily from inputs to pyramidal cells, as they have relatively large dipole moments [37]; we therefore took the net input to the $E$ sub-population to represent the LFP signal. Fig 1B shows examples of raw simulated LFP traces for different stimulus contrasts. For high enough contrast (including all nonzero contrasts shown), the LFP signal exhibits oscillatory behavior. These oscillations can be studied via their power-spectra (Fig 1E, dots; see Methods). As Fig 1F shows, the peak frequencies of the simulated LFP power-spectra shift to higher frequencies with increasing contrast. The two-population SSN model thus captures the empirically observed contrast dependence of gamma peak frequency (compare Fig 1F with Fig 1I of [1] reproduced here as Fig 1D).

## Linearized approximation

To understand this behavior better, we employed a linearization scheme for calculating the LFP power-spectra. The linearization method allows for faster numerical computation of the LFP power-spectra, without the need to simulate the stochastic system Eq 1. More importantly, the linearized framework allows for analytical approximations and insights, which as we show below, elucidate the mechanism underlying the contrast dependence of the gamma peak. We thus explain this approximation with some detail here (see Methods for further details).

First, we note that the linearized approximation scheme is only meaningful when the noise-free network is in a regime of damped (as opposed to sustained) oscillations that decay to a stable steady state (fixed point) with constant neural activity. While in the absence of noise, damped oscillations are transient, input noise constantly rekindles them (c.f. Fig 1B). When input noise fluctuations are sufficiently fast, noise-driven damped oscillations manifest as a resonance peak in the power spectrum of network activity (as in Fig 1E). Empirically recorded gamma peaks are consistent with such a mechanism and have been modelled as such [27, 29]. Alternatively, the noise-free network can be in a regime of sustained oscillations. Changes in the network's connectivity parameters or stimulus input can lead to a transition, a so-called Hopf bifurcation, between the regimes of damped *vs.* sustained oscillations. With weak noise, sustained oscillations create very sharp peaks in the LFP power spectrum, followed by trailing peaks at subsequent harmonics which are rarely visible in LFP recordings. However, given strong enough noise, sustained oscillations above a Hopf bifurcation can also lead to realistic gamma peaks [28, 30]. Indeed, in the presence of noise, the Hopf bifurcation ceases to be a well-defined sharp transition, and the network behavior just below or just above the transition point —as defined in the noise-free network— can be very similar (we will provide examples of this below, in the context of the retinotopic SSN model). Thus, because of the theoretical and computational benefits of the linearized approximation noted above, we limited our main explorations and analyses to the regime of damped oscillations below the Hopf bifuraction. We briefly examine the regime above the Hopf bifurcation at the end of the Results.

Assuming the network is below the Hopf bifurcation, the linearization scheme proceeds as follows. In any stimulus condition (corresponding to a given $\mathbf{I}_{DC}$), we first find the network's steady state in the absence of noise, by numerically solving the noise-free version of Eq 1 (without linearization). The corresponding fixed-point equations can be simplified if we sum them over $\alpha$, and define $\mathbf{h}_* \equiv \sum_\alpha \mathbf{h}_*^\alpha$ and $\mathbf{I}_{DC} \equiv \sum_\alpha \mathbf{I}_{DC}^\alpha$. We then arrive at the same fixed-point equation for $\mathbf{h}_*$ as in the original SSN [23]:

$$\mathbf{h}_* = WF(\mathbf{h}_*) + \mathbf{I}_{DC}. \tag{3}$$

After numerically finding $\mathbf{h}_*$, we then expand Eq 1 to first order in the noise and noise-drive deviations around the fixed point, $\delta\mathbf{h}_t^\alpha \equiv \mathbf{h}_t^\alpha - \mathbf{h}_*^\alpha$, to obtain

$$\tau_\alpha \frac{d\delta\mathbf{h}_t^\alpha}{dt} = -\delta\mathbf{h}_t^\alpha + \tilde{W}^\alpha \sum_\beta \delta\mathbf{h}_t^\beta + \boldsymbol{\eta}_t^\alpha \tag{4}$$

where we defined

$$\tilde{W}^\alpha \equiv W^\alpha \operatorname{diag}\left(F'(\mathbf{h}_*)\right), \tag{5}$$

where $F'(\mathbf{h}_*)$ denotes the vector of gains (slopes) of the I/O functions of different neurons at the operating point $\mathbf{h}_*$ (see the red tangent lines in Fig 1A inset), and diag constructs a diagonal matrix from the vector. As we explain in the next subsection, the neural gains and their dependence on the operating point rates (themselves dependent on the stimulus $\mathbf{I}_{DC}^\alpha$, via Eq 3) play a crucial role in the contrast dependence of gamma peak frequency.

Technically, the linear approximation is valid for small noise strengths, but we found that for the noise levels that elicited fluctuations with realistic sizes, the approximation was very good. As shown in Fig 1E and 1F, the LFP power-spectra and their peak frequencies obtained using the linear approximation agree very well with those estimated from the direct stochastic simulations of Eq 1. The firing rates of $E$ and $I$ units at the fixed-point solution Eq 3 also provide a very good approximation to their mean steady-state rates, at different contrasts, as obtained from direct stochastic simulations of Eq 1 (Fig 1C). Below, we will thus calculate all power-spectra using the computationally faster noise-free determination of the fixed point, Eq 3, and the linear approximation, Eq 4, instead of stochastic simulations of Eq 1.

## Robustness of the two-population model

To demonstrate that the SSN robustly produces the contrast dependence of gamma peak frequency, we simulated 1000 different instances of the 2-population network with parameters randomly drawn from wide but biologically plausible ranges. The sampled parameters were the weights of the connections between the two units ($E \to I$, $I \to E$) and their self-connections ($E \to E$, $I \to I$), the relative strength of input to the excitatory and inhibitory units, and the NMDA fraction of excitatory synaptic weights.

The parameters were sampled independently except for the enforcement of two inequality constraints which previous work has shown to be necessary for ensuring the network's dynamical stability without strong inhibition domination leading to very weak excitatory activity (see Methods and S1 Table for details). The parameter set was also rejected if the resulting SSN did not reach a stable fixed point for all studied stimulus conditions (this corresponds to our modelling choice to have the SSN in a damped oscillation regime).

The majority of randomly sampled models produced steady-state excitatory and inhibitory firing rates that were within biologically plausible ranges, across all contrasts (Fig 2A and 2D). Furthermore, many two-population networks produced peak frequencies that were in the gamma band (30–80 Hz) for all contrast conditions, though some produced peaks at higher frequencies for the highest contrasts (Fig 2B). The distributions also shift towards higher frequencies with increasing contrast, suggesting that the two-population SSN is indeed able to robustly reproduce the contrast dependence of gamma peak frequency. To demonstrate this more directly, we show the distributions of the changes in peak frequency normalized by the change in contrast in Fig 2E. No sampled network produced a negative change in peak frequency with increasing stimulus contrast. As a further corroboration of our model, we also studied how the width of the gamma peak changed with increasing contrast. While [1, 2] did not quantify changes in their gamma peak width with increasing contrast, their results suggest

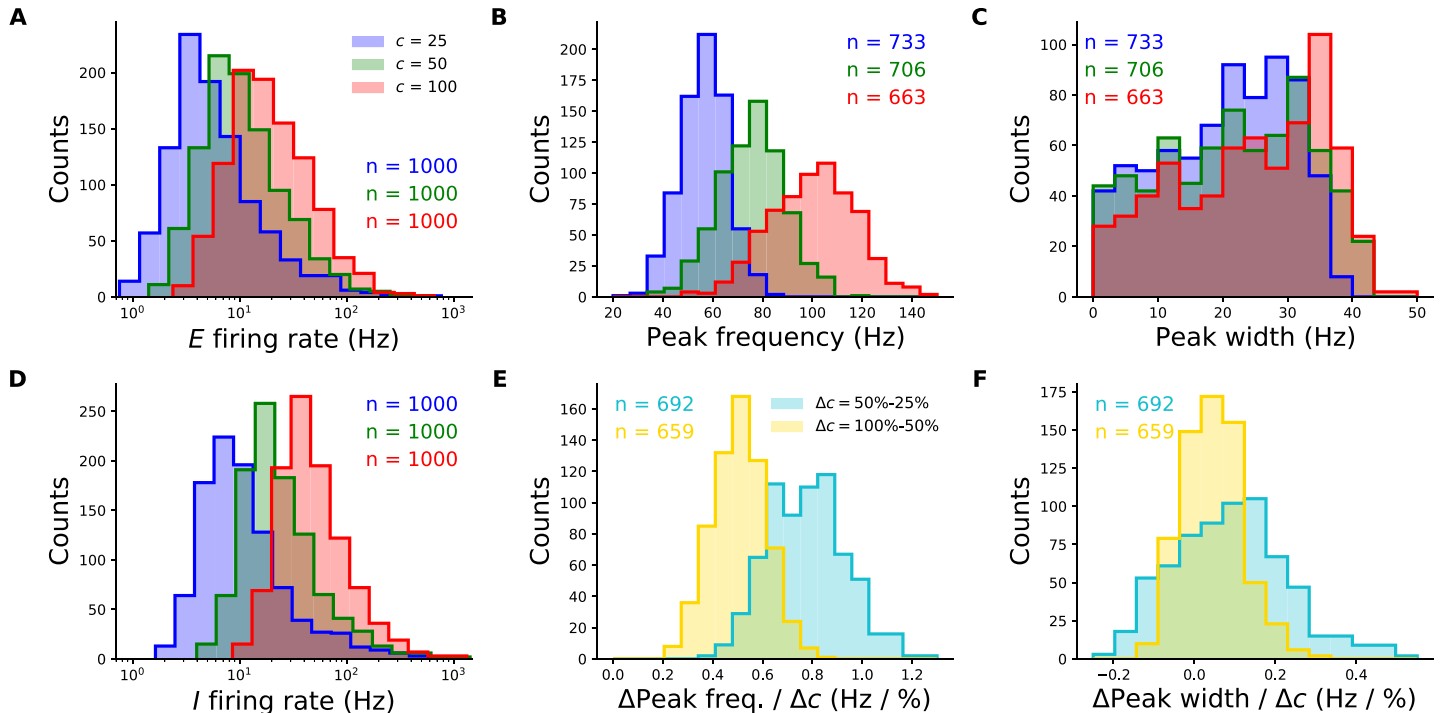

**Fig 2. Robustness of the contrast-dependence of gamma peak frequency to network parameter variations.** One thousand 2-population SSN's were simulated with randomly sampled parameters (but conditioned on producing stable noise-free steady-states), across wide biologically plausible ranges. All histograms show counts of sampled networks; the total numbers (n's) vary across different histograms, as different subsets of network produced the corresponding feature or value in the corresponding condition (*e.g.*, a gamma peak at 50% contrast). **A:** Distributions of the excitatory unit's firing rate in response to 25%, 50%, and 100% contrast stimuli (blue, green, red), plotted on a logarithmic scale. 100% of networks shown across all contrasts. **B:** Distributions of the gamma peak frequencies at different stimulus contrasts. The n's (upper right) give the number of networks with a power spectrum peak above 20 Hz. **C:** Distributions of the gamma peak widths at different stimulus contrasts. **D:** Same as panel A, but for the inhibitory unit. **E:** Distributions of the change in gamma peak-frequency normalized by the change in stimulus contrast, either 25% and 50% (cyan) or 50% and 100% (yellow). **F:** Same as panel E, but for gamma peak-width.

that no significant systematic change in width was observed (Fig 1D). Similarly, in our two-population network, changes in the half-width of the gamma peak are relatively small, and the direction of change can be positive or negative with similar probability (Fig 2F).

## Mechanism underlying the contrast dependence of gamma peak frequency

As we will now show, the SSN sheds light on the mechanism underlying the contrast dependence of the gamma peak, and specifically pins it to the increasing neuronal gain with increasing neuronal activity, due to the expansive, supralinear nature of the neuronal I/O transfer function. This also explains the robustness of the effect to changes in connectivity and external input parameters as demonstrated in the previous subsection.

In the linearized approximation, the LFP power-spectrum (see Eqs 24–26 in Methods) can be expressed in terms of the so-called Jacobian matrix, *i.e.* the matrix of couplings of the dynamical variables, $\delta\mathbf{h}_t^\alpha$, in the linear system Eq 4; thus, for a network of $N$ neurons, the Jacobian is a $3N \times 3N$ matrix, or $6 \times 6$ in the two-population model (see Eq 33 for the explicit form). The existence of damped oscillations and the value of their frequency (which is the resonance frequency manifesting as a peak in the power-spectrum) are in turn determined by the existence of complex eigenvalue of the Jacobian matrix, and the value of their imaginary parts. Previously, the eigenvalues of the Jacobian for a standard *E-I* firing rate network, without

different synaptic current types, were analyzed by [16], and conditions for emergence of damped or sustained oscillations were found. In Methods (see Eigenvalue spectra of rate-based and multi-receptor SSNs in the presence and absence of NMDA), we show that, given the slowness of NMDA receptors relative to gamma timescales, the effect of NMDA receptors on the relevant complex eigenvalues of the Jacobian can be safely ignored, and as a result, only two (out of 6) eigenvalues of the resulting Jacobian can become complex (and thus able to create a gamma peak). Moreover, this pair (which we denote by $\lambda_\pm$) correspond to the two eigenvalues of a standard *E-I* rate model [16] whose *E* and *I* neural time-constants are given, respectively, by the AMPA and GABA decay times:

$$2\lambda_\pm = \gamma_E(\tilde{W}_{EE} - 1) - \gamma_I(\tilde{W}_{II} + 1)$$
$$\pm\sqrt{\left[\gamma_E(\tilde{W}_{EE} - 1) + \gamma_I(\tilde{W}_{II} + 1)\right]^2 - 4\gamma_E\gamma_I\tilde{W}_{EI}\tilde{W}_{IE}} \tag{6}$$

where $\gamma_E = \tau_{\text{AMPA}}^{-1}$ and $\gamma_I = \tau_{\text{GABA}}^{-1}$. Here we defined

$$\tilde{W}_{ab} \equiv W_{ab}F\prime(h_a^*) \qquad (a, b \in \{E, I\}), \tag{7}$$

where $W_{ab} \equiv \sum_\alpha W_{ab}^\alpha$ is the total synaptic weight from unit *b* to unit *a*, and (as in Eq 5) $F'(h_a^*)$ is the gain of unit *a*, *i.e.* the slope of its I/O function, at the operating point set by the stimulus. We refer to $\tilde{W}_{ab}$ as effective synaptic connection weights. Unlike raw synaptic weights, these effective weights are modulated by the neural gains, and thereby by the activity levels in the steady state operating point, which is in turn controlled by the stimulus.

As mentioned, network oscillations emerge when the above eigenvalues are complex (in which case $\lambda_+$ and $\lambda_-$ are complex conjugates). This happens when the expression under the radical in Eq 6 is negative, *i.e.*

$$4\gamma_E\gamma_I\tilde{W}_{EI}\tilde{W}_{IE} > \left[\gamma_E(\tilde{W}_{EE} - 1) + \gamma_I(\tilde{W}_{II} + 1)\right]^2. \tag{8}$$

Qualitatively, the left hand side of the above inequality is a measure of the strength of the effective negative feedback between the *E* and *I* sub-populations, while the right hand size is a measure of the positive feedback in the network (arising from the network's recurrent excitation and disinhibition). Oscillations thus emerge when the negative feedback loop between *E* and *I* is sufficiently strong, in the precise sense of Eq 8.

During spontaneous activity (when the external input is zero or very weak), the rates of both *E* and *I* populations are very small. This means that the spontaneous activity operating point sits near the rectification of the neuronal I/O transfer functions where the neural gains are very low (Fig 1A left). Thus in the spontaneous activity state, the dimensionless effective connections are relatively small. In the limit of $\tilde{W}_{ab} \to 0$, the left hand side of Eq 8 goes to zero, while its right side goes to $(\gamma_I - \gamma_E)^2$ which is generically positive; hence the inequality is not satisfied. This shows that the spontaneous activity state generically does not exhibit oscillations, in agreement with lack of empirical observation of gamma oscillation during spontaneous activity.

On the other hand, when condition Eq 8 does hold, the frequency of the oscillations are given by the imaginary part of the eigenvalues, *i.e.*

$$\text{resonance frequency} =$$
$$\frac{1}{2\pi}\sqrt{\gamma_E\gamma_I\tilde{W}_{EI}\tilde{W}_{IE} - \left[\gamma_E(\tilde{W}_{EE} - 1)/2 + \gamma_I(\tilde{W}_{II} + 1)/2\right]^2} \tag{9}$$

(the division by $2\pi$ is because the eigenvalue imaginary parts give the angular frequency). As

we will discuss further below, the resonance frequency (or approximately the gamma peak frequency Fig 3B) thus depends on the effective connections weights and is thus modulated by the neural gains.

(Note, however, that the *scale* or order of magnitude of this frequency is set by $\gamma_E$ and $\gamma_I$, *i.e.*, by the decay times of AMPA and GABA, as the effective connection weights are dimensionless and cannot determine the dimensionful scale of the gamma frequency; ignoring the "positive feedback" contribution in Eq 9, we find resonance frequency $\propto \sqrt{\gamma_E \gamma_I}/(2\pi)$, which for $\tau_{AMPA} \sim \tau_{GABA} \sim$ 4–6 ms, is on the order of 30–40 Hz).

Eq 9 provides the insight into the contrast dependence of the gamma peak frequency (see Fig 3). As contrasts increase the fixed-point firing rates increase (Fig 1C). Because the SSN I/O transfer function is non-saturating and supralinear, as the rates increase the gains (*i.e.*, the

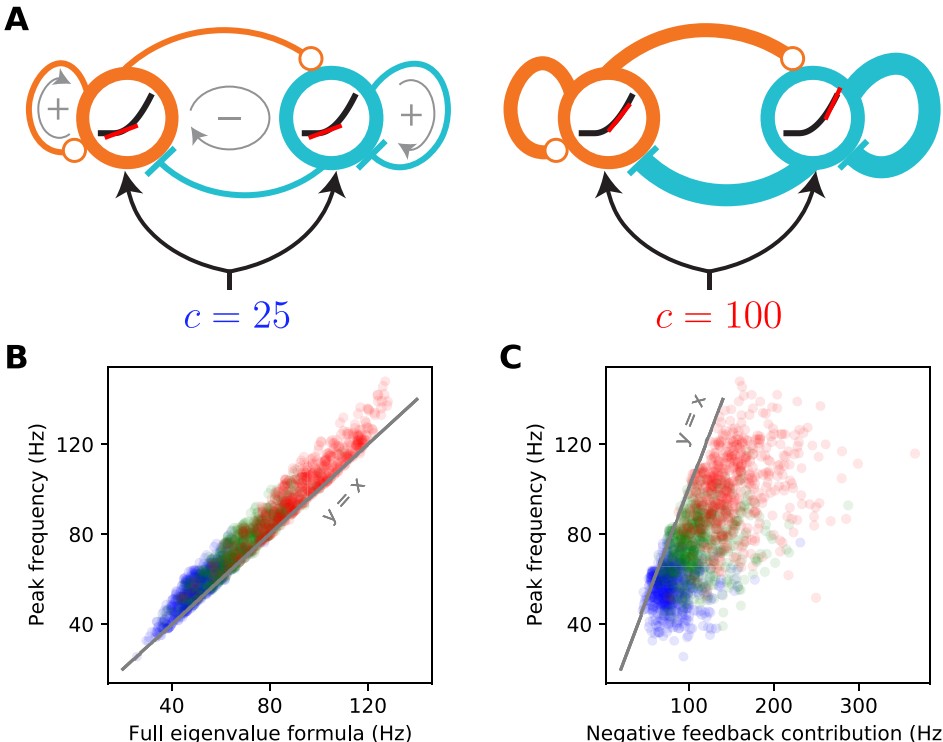

**Fig 3. The supralinear nature of the neural transfer function can explain the contrast dependence of gamma frequency. A:** Schematic diagrams of the 2-population SSN (see Fig 1A) receiving a low (left) or high (right) contrast stimulus. The thickness of connection lines represents the strength of the corresponding effective connection weight, which is the product of the anatomical weight and the input/ouput gain of the presynaptic neuron. The gain is the slope (red line) of the neural supralinear transfer function (black curve), shown inside the circles representing the *E* (orange) and *I* (cyan) units. A resonance frequency exists when the effective "negative feedback" (gray arrow enlcosing a minus sign) dominates the effective "positive feedback" (gray arrows enclosing positive signs), in the sense of the inequality Eq 8. As the stimulus drive (*c*) increases (right panel), the neurons' firing rates at the network's operating point increase. As the transfer function is supralinear, this translates to higher neural gains and stronger effective connections. When a resonance frequency already exists at the lower contrast, this strengthening of effective recurrent connections leads to an increase in the gamma peak frequency, approximately given by the imaginary part of the linearized SSN's complex eigenvalue, Eq 6. **B:** The eigenvalue formula Eq 9 provides an excellent approximation to the gamma peak frequency across sampled networks and contrasts simulated in Fig 2; correlation coefficient = 0.98 ($p < 10^{-6}$), for all data points combined across 25% (blue), 50% (green), and 100% (red) contrasts. **C:** The negative feedback loop contribution to the resonance frequency (Eq 9 with the second term under the square root neglected) overestimates gamma peak frequency but is positively and significantly correlated with it; correlation coefficient = 0.67 ($p < 10^{-6}$).

slope of the I/O transfer functions) of the *E* and *I* cells are also guaranteed to increase (Fig 3A right *vs.* left). The increase in the gains leads in turn to the strengthening of the effective connection weights, Eq 7, and therefore of the network's negative *E-I* feedback loop. When Eq 8 is satisfied, a rough approximation (Fig 3B *vs.* Fig 3C) to the resonance frequency is obtained by ignoring the positive feedback contribution (the second term under the square root in Eq 9). With only the negative feedback contribution retained, it is clear that an increase in neural gains leads to an increase in the resonance frequency (the precise conditions for this to occur are given in Methods, under Eigenvalue spectra of rate-based and multi-receptor SSNs in the presence and absence of NMDA). Thus as contrast increases, we expect the gamma peak in the LFP power-spectrum to move to higher frequencies, due to increasing neural gains and effective connectivity as dictated by the supralinear neural I/O transfer function.

## Retinotopic SSN

We next investigated whether the SSN can account for the locality of contrast dependence of gamma peak frequency, when V1 receives a stimulus with a spatially varying contrast profile. To this end, we expanded our network from two units representing global *E* and *I* populations to many units that are retinotopically organized. We thus model the cortex as a two-dimensional grid that has an *E* and *I* sub-population (corresponding to SSN units) at each grid location, corresponding to a cortical column (Fig 4A).

In the retinotopic SSN, the stimulus input can vary across the network: each column can receive a different input proportional to the contrast within its receptive field. We presented this network with uniform-contrast grating stimuli of various sizes and contrasts (the stimulus in Fig 4A), as well as a Gabor stimulus (Fig 4E), similar to the one used in [1], with a contrast profile that decays smoothly with deviation from the stimulus center according to a Gaussian profile (see Eq 54). Gamma peak frequency shows only a weak dependence on stimulus orientation [2], possibly due to the averaging of LFP over an area larger than the size of orientation minicolumns. To keep our model parsimonious and computationally more tractable, we thus chose the size of our cortical columns to be roughly half the hypercolumn size in Macaque, and neglected the orientation map structure, and the dependence of external inputs and horizontal connections on preferred orientation.

We wish to study the trade-off in this model between capturing surround suppression of firing rates and capturing the local dependence of gamma peak frequency, and asked whether parameter choices exist for which the model can capture both of these effects. In particular, we studied the effect of the spatial profile of the horizontal recurrent connections between and within different cortical columns on this trade-off. In one extreme, we can consider a network in which long-range connections between different columns are very weak, and thus cortical columns are weakly interacting and can be approximated by independent two-population networks which were studied above (Figs 1 and 2). In this case, the frequency of the gamma resonance in each column depends only on the gains and activity levels in that column, which are in turn set by the feedforward input to that column, controlled by the local stimulus contrast. Therefore a network with such a connectivity structure would trivially reproduce the local contrast-dependence of gamma peak frequency. However, due to lack of strong inter-columnar interactions, such a network would fail to produce significant surround suppression of firing rates. In the other extreme, inter-columnar strengths are strong and, importantly, have a smooth fall-off (*e.g.* an exponential fall) with growing distance between pre- and post-synaptic columns; this is the case in most cortical network models, including the SSN model of [22] that captures surround suppression and its various contrast-dependencies. However, as we will show below, in such networks, when horizontal connections are strong enough to produce

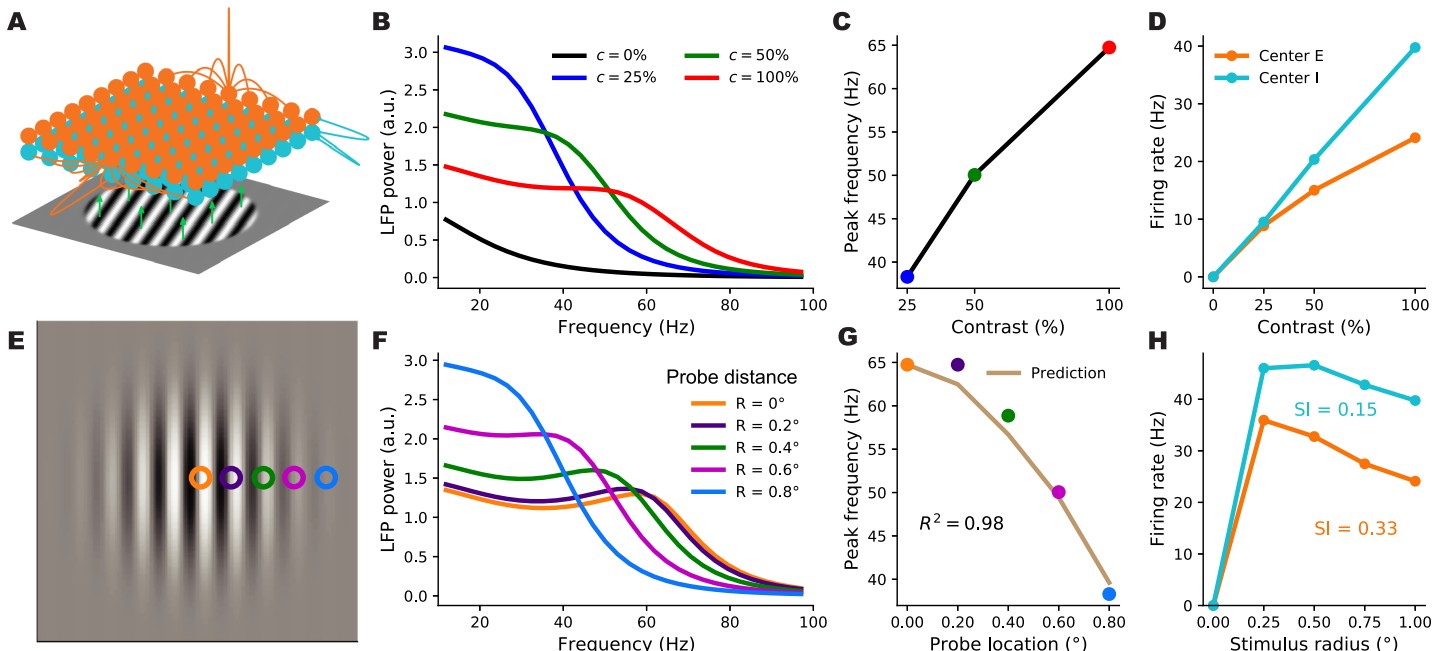

**Fig 4. A retinotopically-structured SSN model of V1, with a boost to local, intra-columnar excitatory connectivity, exhibits a local contrast-dependence of gamma peak frequency, as well as robust surround suppression of firing rates. A:** Schematic of the model's retinotopic grid, horizontal connectivity, and stimulus inputs. Each cortical column has an excitatory and an inhibitory sub-population (orange and cyan balls) which receive feedforward inputs (green arrows) from the visual stimulus, here a grating, according to the column's retinotopic location. Orange lines show horizontal connections projecting from two *E* units; note boost to local connectivity represented by larger central connection. Inhibitory connections (cyan lines) only targeted the same column, to a very good approximation. **B:** LFP power-spectra in the center column evoked by flat gratings of contrasts 25% (blue), 50% (green), and 100% (red). **C:** Gamma peak frequency as a function of flat grating contrast. Note that peaks were defined as local maxima of the relative power-spectrum, *i.e.*, the point-wise ratio of the absolute power-spectrum (as shown in **B**), to the power-spectrum at zero contrast; see Methods. **D:** Firing rate responses of *E* (orange) and *I* (cyan) center sub-populations as a function of grating contrast. **E:** The Gabor stimulus with non-uniform contrast (falling off from center according to a Gaussian). The colored circles show the five different cortical locations (retinotopically mapped to the visual field) probed by the LFP "electrodes". The orange probe was at the center and the distance between adjacent probe locations was 0.2° of visual angle (corresponding to 0.4 mm in V1, the width of the model columns). **F:** LFP power-spectra evoked by the Gabor stimulus at different probe locations (legend shows the probe distances from the Gabor center). **G:** Gamma peak frequency of the power-spectra at increasing distance from the Gabor center. The golden curve is the prediction for peak frequency in the displaced probe location based on the Gabor contrast in that location and the gamma peak frequency obtained in the center location for the flat grating of the same contrast. The predictor's fit to actual Gabor frequencies is very tight ($R^2 = 0.98$), exhibiting local gamma contrast dependence. **H:** Size tuning curves of the center *E* (orange) and *I* (cyan) subpopulations, at full contrast. *E* and *I* firing rates vary non-monotonically with grating size and exhibit surround suppression (suppression indices were 0.33 and 0.15, respectively).

surround suppression, the gamma peak frequency is typically shared across all activated columns, regardless of the spatial contrast profile of the stimulus, and thus this connectivity structure cannot capture the local contrast-dependence of gamma peak frequency. Indeed, as shown below, within this class of networks (*i.e.*, those with a smooth spatial connectivity profile), we did not find connectivity parameters (controlling the range and strength of horizontal connections) for which the network could produce significant surround suppression and yet capture the local contrast-dependence of gamma (see Figs 5 and 6 below).

We then asked whether connectivity structures which involve a sum of strong, spatially smooth long-range connections and an additional boost to local, intra-columnar connectivity could produce both of these effects. In such a structure, the connection strength between two units first undergoes a sharp drop when the distance between the units exceeds the width of a column, and then falls off smoothly over a longer distance, possibly ranging over several columns. This modification can be thought of as adding a local, intra-columnar-only component to a connectivity profile with smooth fall-off. Specifically, we let the excitatory horizontal connections in our model have such a form. Denoting the strength of connection from the unit of

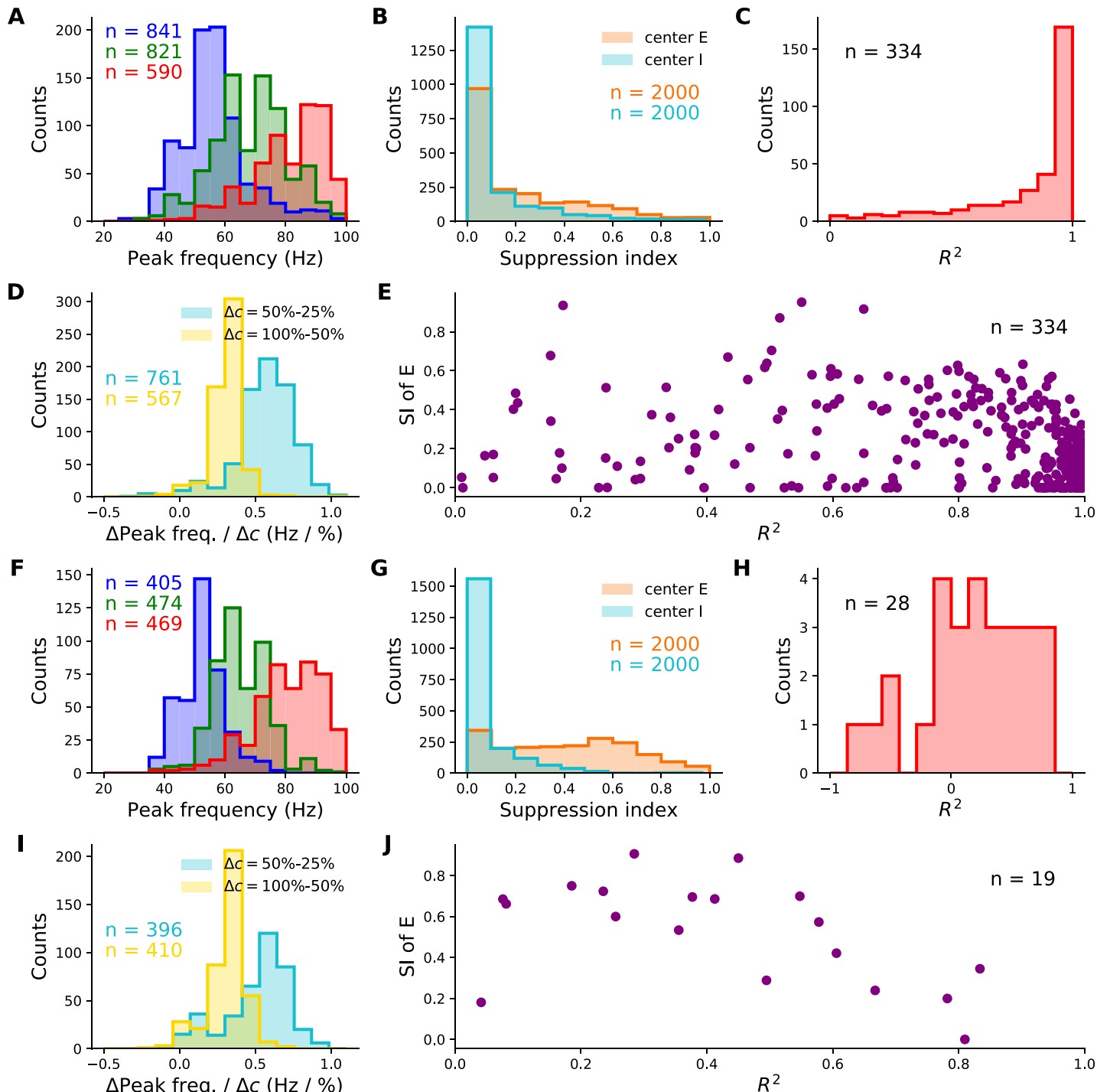

**Fig 5. Behaviour of retinotopic V1 models with and without boosted intra-columnar recurrent excitatory connectivity (columnar *vs.* non-columnar models, respectively) across their parameter space.** We simulated 2000 different networks of each type with parameters (11 in total) randomly sampled across wide, biologically plausible ranges. All histograms show counts of sampled networks; the total number of samples vary across histograms, as only subsets of networks exhibited the corresponding feature with a value in the shown range. Panels A-E and F-J show results for the columnar and non-columnar models, respectively. **A & F:** Distributions of gamma peak frequency, recorded at stimulus center, for different contrasts of the uniform grating (histograms for 25%, 50%, and 100% contrasts in blue, green and red, respectively). **D & I:** Distributions of the change in gamma peak-frequency normalized by the change in grating contrast, changing from 25% to 50% (cyan) or from 50% to 100% (yellow). **B & G:** Distributions of the suppression index for the center $E$ (orange) and $I$ (cyan) sub-populations. **C & H:** The distributions of the coefficient of variation $R^2$, as a measure of the locality of gamma peak contrast dependence. The $R^2$ quantifies the goodness-of-fit of predicted gamma peak frequency based on local Gabor contrast (see Fig 4G showing such a fit in an example network). **E & J:** The joint distribution of $R^2$ and the suppression index of the center $E$ sub-population. Only

a very small minority of sampled non-columnar networks produced $R^2 > -1$ and $R^2 > 0$ to appear in **H** and **J**; hence the small $n$'s, corresponding to 1.4% and 1% of samples, respectively.

type $b$ at location $\mathbf{y}$ to the unit of type $a$ at location $\mathbf{x}$ by $W_{\mathbf{x},a|\mathbf{y},b}$ (with $a, b \in \{E, I\}$), we thus chose:

$$W_{\mathbf{x},a|\mathbf{y},E} \propto \lambda_{a,E}\delta_{\mathbf{x},\mathbf{y}} + (1 - \lambda_{a,E})e^{\frac{-\|\mathbf{x}-\mathbf{y}\|}{\sigma_{a,E}}} \qquad (a \in \{E, I\}),$$  (10)

where $\delta_{\mathbf{x},\mathbf{y}}$ (the local component) is the Kronecker delta: 1 when $\mathbf{x} = \mathbf{y}$ and zero otherwise. The $\lambda_{E,E}$ and $\lambda_{I,E}$ parameters lie between 0 and 1, and interpolate between the two extremes of connectivity structure: for $\lambda_{a,E} = 0$ the horizontal connectivity profile has only one spatial scale and falls off smoothly with distance, while for $\lambda_{a,E} = 1$ connectivity is purely local and intra-columnar. The orange lines in Fig 4A show examples of this connectivity profile. Below we will refer to horizontal excitatory connectivity structures with nonzero (and significant) $\lambda_{a,E}$ as columnar, and to those with $\lambda_{a,E} = 0$ as non-columnar; we will also refer to SSN models with these connectivity types as "columnar" and "non-columnar" models or networks, respectively, for short. As in previous work [22], we chose a smooth Gaussian profile for inhibitory

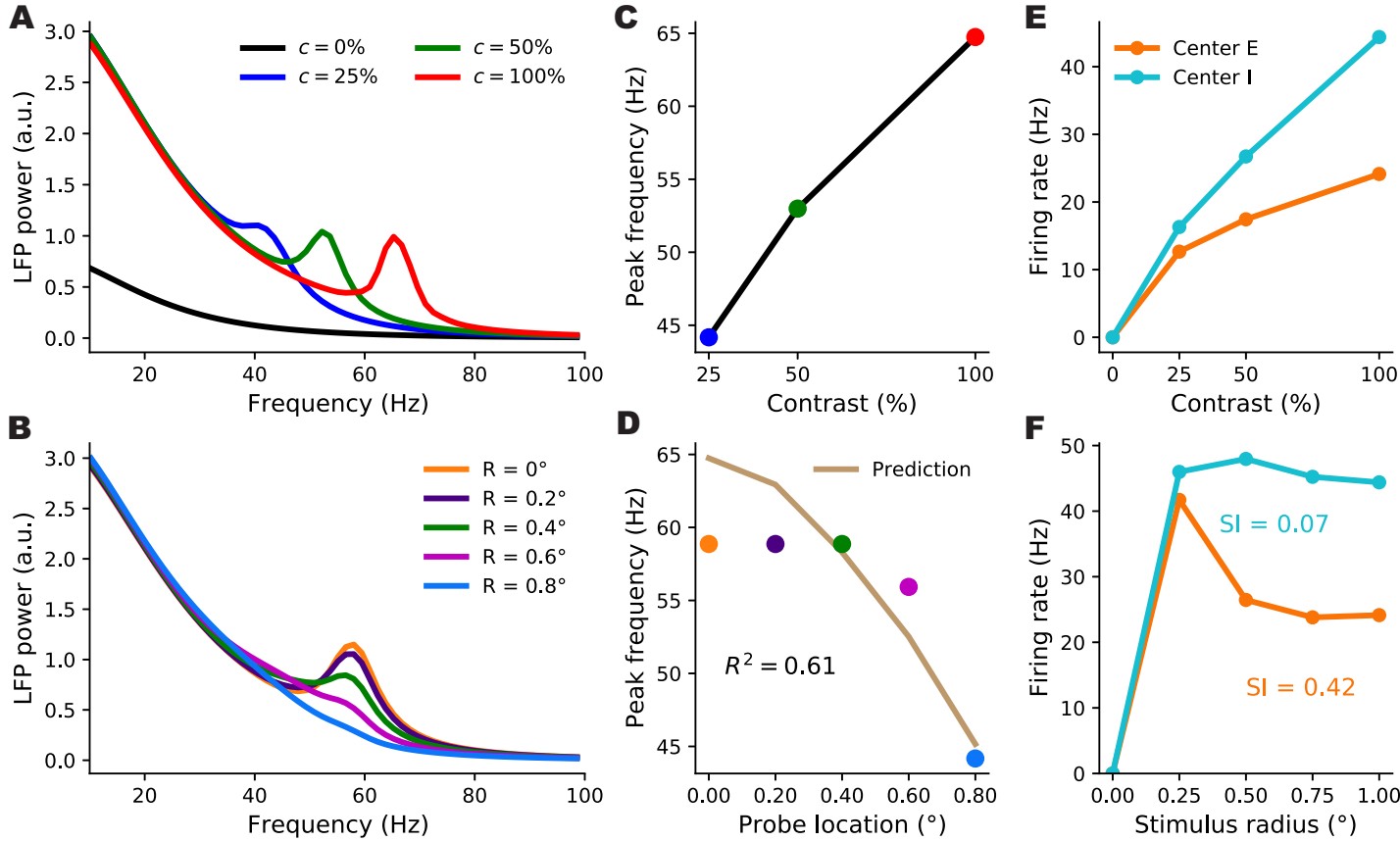

**Fig 6. Gamma peak contrast dependence and surround suppression in an example non-columnar retinotopic SSN without a boost in intra-columnar $E$-connections.** Panel descriptions are the same as for the right three columns in Fig 4. Among 2000 sampled non-columnar networks, this network was the best sample we found in terms of capturing the local contrast dependence of gamma peak frequency (subject to having a nonzero suppression index, and producing realistic non-multiple gamma peaks in the LFP power-spectra which are not unbiologically sharp). However, as seen in panel D, this network only yielded an $R^2$ of 0.61, as an index of locality for the gamma peak's contrast dependence, and gamma peak frequency stayed roughly constant over most of the area covered by the Gabor stimulus (panel B).

connections (see Eq 51), with a relatively short range (see the cyan lines in Fig 4A); thus inhibitory projections essentially only targeted the source column. We also explored networks with longer-range inhibitory connections, and found that our results were not sensitive to this choice (see S1 Fig, to be compared with Fig 5 which is discussed below). Horizontal projections of cortical inhibitory neurons indeed have a shorter range compared to projections of excitatory neurons, whose axons arborize over long distances with a characteristic patchy pattern [39].

In Fig 4B–4H, we show the behavior of firing rates and the gamma peak in an example columnar network (with $\lambda_{E,E} = 0.72$ and $\lambda_{I,E} = 0.70$) in response to different stimuli. We first presented this network with flat gratings of varying sizes and contrasts, and measured the LFP power-spectrum and the firing rate responses of the $E$ and $I$ sub-populations at the "center" column, *i.e.*, at the retinotopic location on which the grating was centered. Firing rates of center $E$ and $I$ both increased with contrast (Fig 4D), and for large enough gratings, we verified that the gamma peak frequency also increases with increasing contrast (Fig 4B and 4C), matching the results of the reduced 2-population model and our previously built intuition. To study surround suppression, we formed the so-called size-tuning curve of the center $E$ and $I$ populations, based on their responses to full-contrast flat gratings of different sizes (Fig 4H). Both $E$ and $I$ responses showed surround suppression: the response first grows but then drops with increasing grating size. The center $E$ sub-population had a suppression index (SI, see Methods for the definition; SI = 0 is no suppression, = 1 is complete suppression) of 0.33 consistent with biologically reported values of suppression indices [15].

To study the locality of contrast dependence, we modeled the experiment of [1], and presented this network with a Gabor stimulus, which has spatially varying contrast with a gaussian profile. We then computed the LFP spectrum at five locations ("columns") of increasing distance from the center of the Gabor stimulus (the colored squares in Fig 4E) with the farthest one lying at 0.8 degrees of visual angle from the Gabor center (compare with the Gabor's $\sigma$ which was 0.5˚ as in [1]). The LFP power-spectra at all locations are shown in Fig 4F. As seen, the gamma peak moves to lower frequencies with increasing distance from the Gabor center, which is accompanied by a decrease in the local contrast (*i.e.*, the contrast of the Gabor stimulus at the receptive field location of the recording site). To quantify the locality of this contrast dependence, we again followed [1], by comparing the actual peak frequency at location **x** with a prediction that solely depended on the local contrast, $c(\mathbf{x})$, of the Gabor stimulus at **x**. The prediction was the peak frequency of gamma recorded at the center location, when the network is presented with a large flat grating of (uniform) contrast equal to $c(\mathbf{x})$. We found that the prediction was in very close agreement with the actual peak frequencies at all distances (Fig 4F). As a measure of the locality of gamma contrast-dependence, we used the corresponding coefficient of determination, $R^2$, which quantifies the agreement between the predicted and actual peak frequencies. (By definition, $R^2 = 1 - \frac{\text{SSE}}{\text{Var}}$, where SSE denotes the sum of squared differences between the predicted and the actual gamma peak frequencies, and Var denotes the variance of the latter; $R^2$ is thus bounded above by 1, which is attained when the prediction perfectly matches the actual data). In the example shown in Fig 4F, we found $R^2 = 0.98$.

To investigate whether the above behavior did or did not require fine tuning of network parameters, we simulated 2000 networks with randomly picked parameters. There were 11 parameters in total, characterising the strengths and ranges of horizontal connections, including $\lambda_{a,E}$, the strength of feedforward connections, and the ratio of NMDA to AMPA in recurrent excitatory synapses. These parameters were were picked randomly and independently (except for the enforcement of three inequality constraints, similar to the samplings of the

two-population model in Fig 2) across wide ranges of values consistent with biological estimates; see Methods and S1 Table for details. Again, similar to the sampling of the two-population model, sampled networks which failed to reach a stable steady-state in any stimulus condition were discarded.

The vast majority of sampled networks produced biologically plausible center excitatory and inhibitory firing rates which increased with increasing contrast (*E* and *I* firing rates did not exceed 100 Hz in, respectively, 90% and 81% of networks). The majority of networks produced surround suppression in excitatory (70% of samples) and inhibitory (53% of samples) populations, with many networks yielding strong suppression in both populations (Fig 5B). In addition, in response to grating stimuli (with uniform contrast profile), many networks produced gamma-band peaks in the LFP power spectrum that moved upward in frequency with increasing contrast (Fig 5A and 5D; the number of networks yielding gamma peak for each nonzero contrast is denoted in panel A). Finally, for each network, we again quantified the locality of contrast dependence of peak frequency, using the $R^2$ coefficient for the match between peak frequencies obtained at different recording locations on the Gabor stimulus, and their predictions based on the local Gabor contrast and the peak frequency obtained using the flat grating with that contrast. A sizable fraction of networks resulted in a high $R^2$ signifying local contrast-dependence of gamma peak frequency (Fig 5C). Moreover, many of these networks exhibited strong surround suppression as well (Fig 5E). Sixty six networks (4.2% of samples) yielded an $R^2 > 0.8$ and $SI_E > 0.25$.

In sum, the columnar model, which emphasizes the intra-columnar excitatory connectivity (Eq 10), can robustly exhibit strong surround suppression in conjunction with gamma peak frequencies controlled by the local contrast, as observed empirically, without requiring a fine tuning of parameters.

## Retinotopic SSNs with non-columnar excitatory connectivity do not account for local contrast dependence of gamma frequency

To further show the importance of a boost in intra-columnar excitatory connectivity for obtaining local contrast dependence despite strong surround suppression, we next sampled retinotopic SSN models without this structure in horizontal connections ("non-columnar" model). In these models horizontal excitatory connections fall off smoothly over distance between the source and target columns (corresponding to $\lambda_{a,E} = 0$ in the notation of Eq 10). We found that while many sampled non-columnar models exhibited strong surround suppression (on average stronger than in the sampled columnar models) none of the sampled models exhibited gamma peak frequencies with sufficiently local contrast-dependence.

The sampled non-columnar networks robustly exhibited surround suppression which was strong in a large fraction of these networks, and, especially in excitatory units, was on average stronger than in the sampled columnar models (Fig 5G *vs*. Fig 5B). Many networks produced LFP power-spectrum peaks in the gamma band with frequency increasing with contrast (Fig 5F and 5I), but sampled networks with these properties were about half as common as in the case of the columnar model (Fig 5A and 5D). Moreover, when presented with the Gabor stimulus, we found that the contrast dependence of gamma peak frequency in the vast majority of non-columnar networks was far from local, and the same peak frequency was shared across most of the retinotopic region stimulated by the Gabor stimulus (see the power spectra of an example sampled non-columnar network in Fig 6A and 6B). Only 1% of sampled non-columnar networks exhibited a positive $R^2$ (our measure of local contrast dependence), compared to 15% of columnar networks (Fig 5H and 5J). Only 4 non-columnar networks (out of 2000 sampled networks) exhibited an $R^2 > 0.6$ in conjunction with any degree of surround suppression

(positive suppression index) of $E$ firing rates (Fig 5J). However, three of these networks which had the highest $R^2$'s exhibited non-biological gamma-band power spectra, featuring either multiple gamma peaks or unrealistically sharp ones. The fourth network produced realistic gamma peaks and achieved an $R^2 = 0.61$; the LFP power spectra, gamma frequency contrast dependence, and size tuning curves for this example network are shown in Fig 6.

We conclude that the non-columnar model class cannot robustly exhibit both surround suppression of firing rates and local contrast dependence of gamma peak frequency.

## Contrast dependence of gamma frequency above the Hopf bifurcation

As mentioned above, under Linearized approximation, and for reasons discussed there, we have heretofore explored the behaviour of the SSN in a regime of noise-driven damped oscillations, which, in the absence of noise, would decay to a fixed point of neural activity. Changes in connectivity or stimulus parameters can nevertheless lead to a Hopf bifurcation (as defined in the noise-free network), namely a transition to a regime of sustained oscillations. In the presence of noise, however, this regime change ceases to be a well-defined and sharp transition. Therefore, given sufficiently strong noise, we expect, on theoretical grounds, that the contrast dependence of gamma oscillations would be qualitatively similar just below or just above the Hopf bifurcation (also note that the same network could be above or below this bifurcation in different stimulus conditions). We directly verified this prediction in the two example networks presented in Figs 4 and 6. Theoretically we expect that a sufficient increase in the strength of recurrent excitation (*i.e.*, the strength of the $E$ to $E$ connections) would result in a Hopf bifurcation. As shown in S2 Fig, we first mapped out this Hopf bifurcation, as we increase the strength of $E$ to $E$ connections, in the noise-free versions of the two, columnar and non-columnar, example networks. We then simulated (as opposed to using the linearized approximation) the two noise-driven networks at a value of recurrent $E$ to $E$ connections above the Hopf bifurcation, and calculated the LFP power spectra in these networks in the different stimulus conditions considered above. The results are presented in S3 and S4 Figs. A comparison of those figures with the corresponding panels in Figs 4 and 6, respectively, shows that the qualitative behavior is indeed very similar in the versions of each network above the Hopf bifurcation (*i.e.*, with connectivity parameters for which the networks exhibit sustained periodic oscillations in the absence of input noise). In particular, the gamma peak frequencies of the columnar example model (see S3 Fig panel E) still exhibits a local contrast-dependence (albeit with a decreased $R^2$ compared to Fig 4G), while the non-columnar model (see S4 Fig panel E) exhibits a virtually constant gamma peak frequency at different locations, despite the stimulus contrast varying strongly over space.

## Mechanism underlying the local contrast dependence of gamma peak frequency in the columnar SSN

We can understand the mechanism underlying the local contrast dependence of gamma frequency in the columnar model, and its failure in the non-columnar model, by looking at the spatial profile of the normal oscillatory modes of these networks. Normal modes are the eigenvectors of the Jacobian matrix, the effective connectivity matrix of the linearized network introduced before Eq 6, and normal *oscillatory* modes are the Jacobian eigenvectors with complex eigenvalues. (As described above, the oscillation frequency is given by the imaginary part of the eigenvalue, and thus we are particularly interested in the eigenvectors of eigenvalues with imaginary part in the gamma band). The Jacobian is dependent on the operating point of the linearization, which is in turn set by the stimulus. The relevant stimulus condition for us is the Gabor stimulus, or more generally a stimulus with non-uniform contrast.

As we discussed above, gamma peak frequency would be trivially determined by local contrast in a network with only local connectivity (corresponding to $\lambda_{aE} = 1$ in our model) and disconnected cortical columns. The disconnected columns act like the 2-population model of the first part, and can oscillate independently of other columns by a frequency set by the operating point of that column, which is in turn set by the stimulus input to that column and thus the local contrast. In such a network, all Jacobian eigenvectors are completely localized spatially at a single column. Since the mode is localized, its eigenvalue and hence its natural frequency, are entirely determined by the stimulus contrast over that column.

By contrast, in the model with long-range connections and $\lambda_{aE} = 0$ (the non-columnar model) the eigenvectors can spatially cover a large region of retinotopic space, and lead to coherent and synchronous oscillations at the same frequency (set by that mode's eigenvalue) across many columns. To see this, consider the case of a stimulus with uniform contrast. Such a stimulus does not break the network's translational symmetry (since recurrent connections do not care about absolute location, and only depend on relative distance of pre- and post-synaptic columns). Due to this symmetry all eigenmodes are completely delocalized and have (sinusoidal) plane wave spatial profiles extending over the entire retinotopic space. A non-uniform stimulus does break the translational symmetry and leads to relative localization of eigenvectors. However, the scale of this localization is set by the scale over which the stimulus contrast varies appreciably. If this variation is smooth, the eigenvectors can still cover a large region. In the case of the Gabor stimulus, the $\sigma$ of the Gaussian profile of this stimulus sets this length scale. Thus eigenvectors tend to cover much of the space covered by the stimulus. This can be seen in Fig 7D showing oscillatory eigenvectors of an example non-columnar network, for modes which maximally contribute to the gamma peaks recorded at different locations. When such an eigenvector has a complex oscillatory eigenvalue (which is sufficiently close to the imaginary axis so that its oscillations are not strongly damped), it will give rise to coherent gamma oscillations across the area covered by the stimulus, at the *same* frequency set by the mode's eigenvalue. The corresponding peak will thus appear at the same frequency in the power-spectra of the LFP recorded across this space, despite smooth variations in local contrast (Fig 7C top). This mechanism thus breaks the local contrast dependence of peak frequency.

Our columnar model, via its parameters $\lambda_{aE}$, interpolates between disconnected networks and the kind of network just discussed. With a sufficient boost of intra-columnar connectivity (*i.e.*, with sufficiently large $\lambda_{aE}$) the eigenvectors of this model become approximately localized, not to single columns, but to a small number of columns receiving similar stimulus contrasts. This is shown in an example columnar network in Fig 7B, showing different Jacobian eigenvectors (all for the Gabor stimulus condition) which are approximately localized at different locations. Even those eigenvectors that are relatively more spread, tend to extend over rings encircling the Gabor's center, and thus cover an area receiving the same contrast. The eigenvalue and natural frequency of such modes is thus largely controlled by that contrast value: modes that do not extend to a given location do not contribute to the power-spectrum recorded at that location, while modes that are localized nearby only "see" the local contrast.

This observation further explains the typical shape of the eigenvalue spectrum observed in columnar networks. As seen in Fig 7A (bottom) the eigenvalue spectrum consists of a near-continuum of eigenvalues extending along the imaginary axis. Eigenvalues with higher (lower) imaginary parts (*i.e.*, the corresponding modes' natural frequency) have eigenvectors localized at regions of higher (lower) contrast. In this way, eigenvalues with different imaginary parts roughly correspond to different locations that have different local contrasts, with imaginary part decreasing with local contrast. By contrast, oscillatory eigenvalues in non-columnar networks, especially eigenvalues near the imaginary axis within the gamma band, tend to be

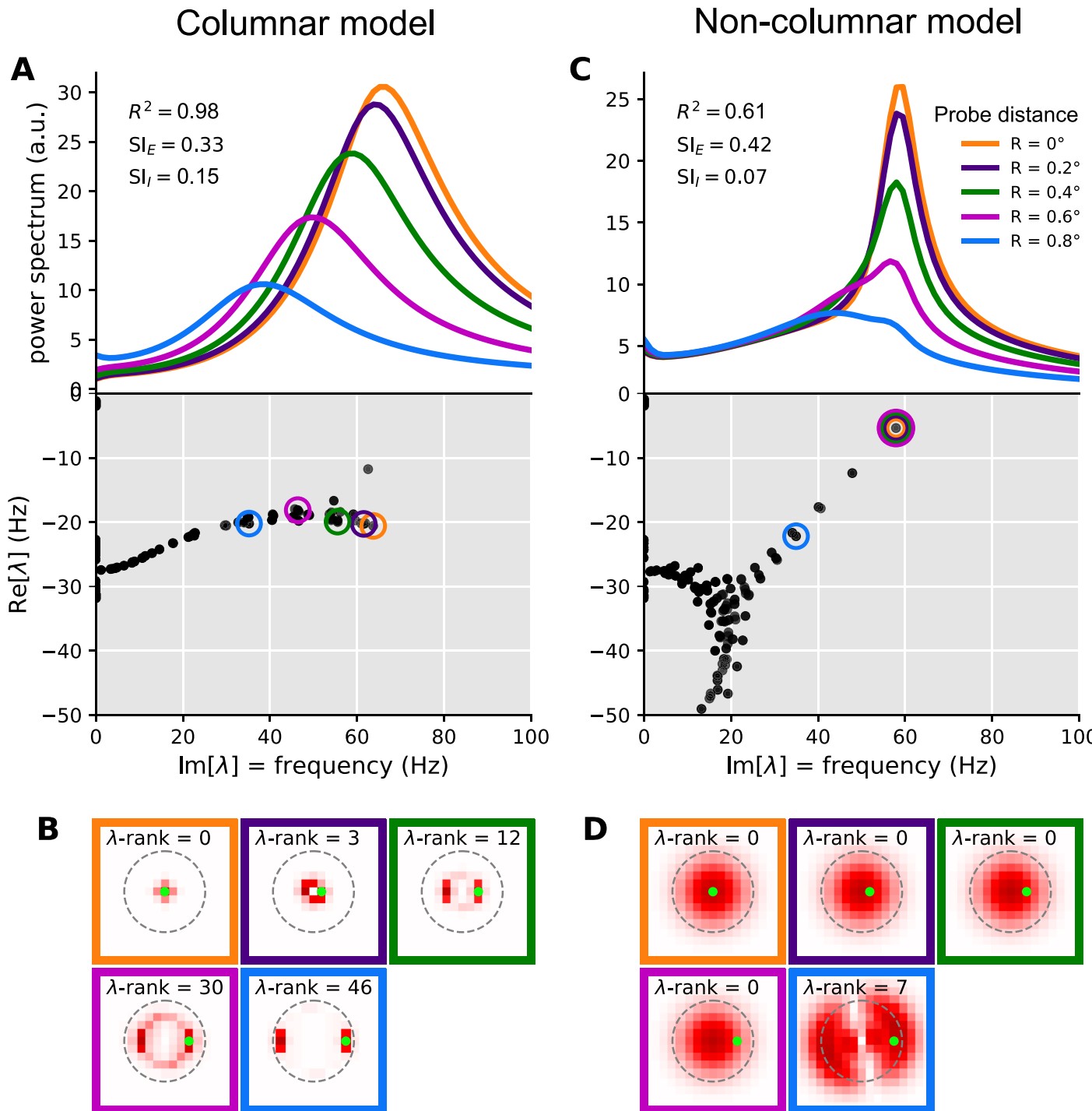

**Fig 7. Mechanism of local contrast dependence in the retinotopic SSN with columnar structure and its failure in the non-columnar model.** The left and right columns correspond to the columnar (retinotopic SSN with boosted local, intra-columnar excitatory connectivity) and non-columnar (retinotopic SSN without the boost in local connectivity) example networks in Figs 4 and 6, respectively. **A:** Top: relative LFP power spectra recorded at different probe locations in the Gabor stimulus condition (see Fig 4E; same colors are used here to denote the different LFP probes). Relative power spectrum is the pointwise ratio of the evoked power spectrum (evoked by the Gabor stimulus) to the spontaneous power spectrum in the absence of visual stimulus (the absolute power spectrum for the same conditions was given in Fig 4F). Bottom: the eigenvalue spectrum in the complex plane, with real and imaginary axis exchanged so that the imaginary axis aligns with the frequency axis on top (eigenvalues are also scaled by $1/(2\pi)$ to correspond to non-angular frequency). The eigenvalues were weighted separately for each probe, according to Eq 11, and the eigenvalue with the highest weight was circled with the probe's color (see Fig 4E). This eigenvalue contributes the strongest peak to the power spectrum at that probe's location. **B:** Each sub-panel corresponds to one of the probe locations (as indicated by the frame color), and plots the absolute value of the highest-weight eigenvector (more precisely, the function $|\mathcal{R}_a(\mathbf{x})|$ defined in Eq 37) over cortical space. Thus, this is the eigenvector corresponding to the circled

eigenvalue in panel A, bottom. The λ-rank in each sub-panel is the order (counting from 0) of the eigenvalue according to decreasing imaginary part, which is the eigen-mode's natural frequency. The green dot in each sub-panel shows the location of the LFP probe. **C-D:** Same as A and B, but for the retinotopic SSN model with no columnar structure.

isolated (Fig 7C bottom shows this in an example non-columnar network); the corresponding mode can not be associated with a given location or contrast. Indeed since the eigenvectors of these modes extend over many columns receiving varying contrasts, the mode's eigenvalue and oscillation frequency are not determined by stimulus contrast at any single column, but rather by the entire spatial profile of contrast, in a complex manner.

The above qualitative discussion can be made quantitative using the linearized approximation. In Methods (see Eqs 39 and 41), we derive an expression for the power-spectrum at a location as a sum of individual contributions by different eigenmodes. (The sum also includes terms that are contributions of different *pairs* of modes, which, depending on whether the two modes interfere constructively or destructively, can be positive or negative —see Eq 43; we did not take into account pair contributions in weighting of eigenmodes). As show in Eq 41, mode $a$, with eigenvalue $\lambda_a$, contributes a peak to the power spectrum located at frequency given by the imaginary part of $\lambda_a$ (the mode's natural frequency). The half-width of the peak is given by minus the real part of the eigenvalue, which we denote by $\gamma_a$. Finally, the peak amplitude is proportional to

$$\frac{|\mathcal{R}_a(\mathbf{x})|^2}{\gamma_a^2}, \tag{11}$$

where $\mathcal{R}_a(\mathbf{x})$ is the component of the mode's right eigenvector at column $\mathbf{x}$'s $E$ population (after summing the components corresponding to different synaptic receptors; see Eq 37). Thus this peak only leaves an imprint on the LFP power-spectrum in locations where this eigenvector has appreciable components. The amplitude is also inversely related to the square of the peak's half-width, $\gamma_a$, which by definition measures the distance between the eigenvalue and the imaginary axis. Thus eigenvalues closer to this axis produce stronger and sharper peaks, which appear in the LFP spectrum probed at location $\mathbf{x}$, only if the corresponding right eigenvector has strong (excitatory) components there. In Fig 7, separately for each of the five LFP probe locations on the Gabor stimulus, we picked the mode with the highest amplitude as defined in Eq 11. The corresponding eigenvalue is circled in the eigenvalue spectrum plots (bottom plots in Fig 7A and 7C for the columnar and non-columnar example models, respectively). In Fig 7B and 7D we then plot the corresponding eigenvectors; more precisely, we have plotted $|\mathcal{R}_a(\mathbf{x})|$ which, according to Eq 11, controls the strength of a mode's contribution at different locations $\mathbf{x}$. As observed, in the non-columnar model, the eigenvectors spread over the entire region covered by the Gabor stimulus. Thus the *same* mode makes the strongest contribution to the LFP spectra (shown in Fig 7A and 7C, top plots) at all probe locations, except for the one that is farthest from the Gabor center. Since this best mode is shared across the first four probe locations, its fixed eigenvalue (Fig 7C, bottom plot) determines the location of the power-spectrum peak (Fig 7C, top plot) in all but the farthest probe location.

By contrast, in the columnar model, eigenvectors cover a considerably smaller area within which contrast varies little. As a result, each mode only affects the LFP power spectrum locally, and when the probe moves, the best mode changes quickly, as if the best eigenvector "moves" with the probe (Fig 7B). In turn, the corresponding best eigenvalues also move to lower frequencies along the imaginary axis (Fig 7A, top), as the probe moves farther from the Gabor center, according to the local contrast "seen" by their eigenvector.

Inter-columnar projections in the columnar model are nevertheless sufficiently strong to be able to give rise to strong surround suppression, as evidenced above. It is also worth noting that for large gratings that give rise to surround suppression, the contrast is uniform over a broad area, in which case even the eigenvectors of the columnar model tend to cover a broad area (mathematically, this is because when stimulated with a uniform stimulus, the columnar model also has approximate translational invariance, and therefore its eigenvectors tend towards delocalized approximate plane waves).

In summary, the columnar model can balance the requirements for locality of gamma contrast dependence and strong surround suppression, because of the intermediate spatial spread of its eigenvectors, which tend to cover relatively small areas with roughly uniform contrast.

## Discussion

In this work we have shown that the expanded SSN is able to robustly display the contrast dependence of gamma peak frequency in both a two-population and a retinotopic network. The retinotopic model successfully balances the trade-off in horizontal connection strength such that both the local contrast dependence of the gamma peak frequency and the surround suppression of firing rates are observed robustly. In order to capture gamma oscillations using the SSN, we expanded the model beyond an E-I network to a varied synaptic network model. Crucially, the SSN account sheds light on the mechanism underlying the contrast dependence of gamma peak frequency and points to the key role of the non-saturating and expansive neural transfer function, observed empirically [40, 41], in giving rise to this effect.

Finding the power-spectra using the linearization to Eq 1, helped us make analytic simplifications. From these simplifications, we gained insights on how the SSN captures the gamma contrast dependence. As contrast increases, firing rates increase, which, due to the supralinear neural transfer function of SSN, lead to increasing neural gains. This in turn strengthens effective connectivity, leading to faster oscillations. Moreover, by finding the power-spectra via linearization, we were able to rapidly compute power-spectra which allowed for extensive explorations of the model's parameter space. Nevertheless, we also explored the behavior of LFP power spectra in our example networks using direct simulations of the networks' stochastic dynamics. These simulations validated the accuracy of the linearized approximation below the Hopf bifurcation (*i.e.*, transition to a regime of sustained periodic oscillations in the noise-free network), and also showed that the contrast-dependence of gamma frequency behaves similarly above and below the Hopf bifurcation (which could occur in the same network in different stimulus conditions), as expected on theoretical grounds.

In this work, for simplicity, we assumed an instantaneous I/O function between net synaptic input ($\sum_\beta \mathbf{h}_\beta$) and the output rate. This is based on the approximations discussed in [36], which is valid when the fast synaptic filtering time-constants ($\tau_{AMPA}$ and $\tau_{GABA}$) are much smaller than the neuronal membrane time-constants. However, our framework can easily be generalized beyond this approximation by using the full neuronal linear response filter obtained from the Fokker-Planck treatment of [36]. The main change due to such a modification would be to render the neural gains frequency dependent. We expect this dependence to be weak because we are in the regime of fast synaptic filtering as compared to the neuronal membrane time constant, and so we expect the transfer function to be approximately instantaneous [36]. Therefore we do not expect that including the full neuronal linear response filter would change our qualitative results.

As we have shown, retinotopic SSN networks account for the co-existence of the local contrast dependence of gamma peak frequency and strong surround suppression by balancing long-range horizontal connection strength that decreases exponentially with distance with an

additional strengthening of very local excitatory connections. One possible scenario is that the additional local connection strength is needed to compensate for the possible effects of the model's coarse retinotopic grid which might have distorted the functional effects of a connectivity profile with a smooth, single-scale fall-off, without a local boost. Alternatively, this may represent a measurable increase in connection strength above an exponential function of distance at short distances, below $\sim 200 - 400\,\mu$m. Indeed, it has been previously noted that anatomical findings on the spatial profile of horizontal connections in the macaque cortex point to such a mixture of short-range or local and long-range connections, with the local component not extending beyond 400 $\mu$m (the size of our model's columns) [42].

It is notable that, in modeling surround suppression in V1, [43] also found that they needed to increase the central weight strengths, relative to an exponential fall-off of strength on a grid, for their SSN model to account for two other observations. These observations were the decrease in inhibition received by a cell when it is surround suppressed [44, 45], and the fact that the strongest surround suppression occurs when surround orientation matches center orientation, even if the center orientation is not the cell's preferred orientation [46, 47]. Other phenomena they addressed could be explained by their SSN model with or without this extra local strength. We believe that other visual cortical phenomena previously addressed with the SSN model [22, 34, 45, 48] would not be affected by such boosting, as the mechanisms inferred behind them appear independent of these connectivity details.

Recently some evidence of enhanced connectivity at very short distances ($\sim 20\,\mu$m) has been found in mouse V1 [49]. Optogenetic stimulation of ten cells found excitation of nearby cells only at such short distances from one of the stimulated cells, with suppression at longer distances. In a model, this required an extra component of connectivity that decreased with distance on a very short length scale, in addition to one with a longer length scale. As they point out, such extra local strength might account for the observation that preferred orientations in mouse visual cortex are correlated on a similar very short length scale [50]. It will be interesting to see if evidence of such a short-length-scale component of connectivity is evident in monkeys, where the local contrast dependence of gamma was measured, and conversely to see if mice show similar local contrast dependence of gamma.

## Methods

### Stabilized supralinear network (SSN) with different synaptic receptor types

In its original form, the Stabilized Supralinear Network (SSN) is a firing rate network of excitatory and inhibitory neurons that have a supralinear rectified power-law input/output (I/O) transfer function:

$$F(h) = k[h]_+^n \tag{12}$$

where $n > 1$ and $[h]_+ \equiv \max(0, h)$ denotes rectification of $h$. The dynamics can either be formulated in terms of the inputs to the units [34] or in terms of their output firing rates [22, 23]. Here we adopt the former case for which the dynamical state of the network, in a network of $N$ neurons, is given by the $N$-dimensional vector of inputs $\mathbf{h}_t$, which evolves according to the dynamical system

$$\mathrm{T}\frac{d\mathbf{h}_t}{dt} + \mathbf{h}_t = WF(\mathbf{h}_t) + \mathbf{I}_t. \tag{13}$$

Here, $\mathbf{I}$ is the external input vector, $\mathrm{T} = \mathrm{diag}(\tau_1, \ldots, \tau_N)$ is the diagonal matrix of synaptic time constants, and $F$ acts element-wise. Finally, $W$ is the $N \times N$ matrix of recurrent connection weights between the units in the network. This connectivty matrix observes Dale's law,

meaning the sign of the weight does not change over columns. If we order neurons such that excitatory neurons appear first and inhibitory neurons second, this matrix takes the form

$$W = \begin{pmatrix} W_{EE} & -W_{EI} \\ W_{IE} & -W_{II} \end{pmatrix} \tag{14}$$

where $W_{XY}$ ($X, Y \in [E, I]$) have non-negative elements.

The above model does not take into account the distinct dynamics of currents through different synaptic receptor channels: AMPA, GABA$_A$ (henceforth GABA), and NMDA. Only the fast receptors, AMPA and GABA, have timescales relevant to gamma band oscillations. These receptors have very fast rise times (on the order of 1 millisecond), which correspond to frequencies much higher than the gamma band. We therefore ignored the rise times of all receptors. The slow decay time of the NMDA makes the portion of fluctuating excitatory inputs filtered by this receptor to have a negligible contribution to power within the gamma band. (A nonzero NMDA rise time would not alter this conclusion. Therefore for simplicity and uniformity we also neglected the rise time of NMDA receptors in our model). The reason we nevertheless include NMDA in the model is that the ratio of NMDA to AMPA connectivity (see Eqs 16 and 17 below) controls the portion of excitatory connection strengths that contributes to gamma oscillations (see Eigenvalue spectra of rate-based and multi-receptor SSNs in the presence and absence of NMDA in Methods for an analytic exposition of this point).

With these assumptions, upon arrival of an action potential in a pre-synaptic terminal at time $t = 0$, the post-synaptic current through receptor channel $\alpha$ ($\alpha \in \{A = \text{AMPA}, G = \text{GABA}, N = \text{NMDA}\}$) with decay time $\tau_\alpha$ is given by $w_\alpha \frac{\theta(t)}{\tau_\alpha} e^{-t/\tau_\alpha}$ where $\theta(t)$ is the Heaviside step function and $w_\alpha$ is the contribution of receptor $\alpha$ to the synaptic weight. This is the impulse response solution to the differential equation $\tau_\alpha \frac{dh_\alpha}{dt} + h_\alpha = w\delta(t)$, where $\delta(t)$ is the Dirac delta representing the spike at time $t = 0$. In the mean-field firing rate treatment, the delta function is averaged and is replaced by a smooth rate function $r(t)$. Extending this to cover post-synaptic currents from all synapses into all neurons we obtain the equation

$$\tau_\alpha \frac{d\mathbf{h}_t^\alpha}{dt} + \mathbf{h}_t^\alpha = W^\alpha \boldsymbol{r}_t \tag{15}$$

where $\boldsymbol{r}_t$ and $\mathbf{h}^\alpha$ are $N$-dimensional vectors of the neurons' firings rates and input currents of type $\alpha$, respectively, and $W^\alpha$ are $N \times N$ matrices containing the contribution of receptor type $\alpha$ to the recurrent synaptic weights. If we add an external input to the right side (before filtering by the synaptic receptors), we obtain Eq 1. Since AMPA and NMDA only contribute to excitatory synapses, and GABA only to inhibitory ones, in general the $W^\alpha$ have the following block structure

$$\begin{aligned} W^{\text{A}} &= \begin{pmatrix} W_{EE}^{\text{A}} & 0 \\ W_{IE}^{\text{A}} & 0 \end{pmatrix}, \quad W^{\text{N}} = \begin{pmatrix} W_{EE}^{\text{N}} & 0 \\ W_{IE}^{\text{N}} & 0 \end{pmatrix}, \\ W^{\text{G}} &= \begin{pmatrix} 0 & -W_{EI}^{\text{G}} \\ 0 & -W_{II}^{\text{G}} \end{pmatrix}. \end{aligned} \tag{16}$$

For simplicity, we further assumed that the fraction of NMDA and AMPA is the same in all excitatory synapses. In this case all $W^\alpha$ can be written in terms of the four blocks of the full

connectivity matrix $W \equiv \Sigma_\alpha W^\alpha$, introduced in Eq 14:

$$
\begin{aligned}
W^{\mathrm{N}} &= \frac{\rho_{\mathrm{N}}}{1-\rho_{\mathrm{N}}} W^{\mathrm{A}} = \rho_{\mathrm{N}} \begin{pmatrix} W_{EE} & 0 \\ W_{IE} & 0 \end{pmatrix}, \\
W^{\mathrm{G}} &= \begin{pmatrix} 0 & -W_{EI} \\ 0 & -W_{II} \end{pmatrix}.
\end{aligned}
\tag{17}
$$

where the scalar $\rho_{\mathrm{N}}$ is the fractional contribution of NMDA to excitatory synaptic weights.

As noted in Results, to close the system of equations for the dynamical variables $\mathbf{h}_t^\alpha$, we have to relate the output rate of a neuron to its total input current, $\mathbf{h}_t^{\mathrm{total}} = \sum_\beta \mathbf{h}_t^\beta$. In general, the relationship between the total input and the firing rate of a neuron, or the mean firing rate of a population of statistically equivalent neurons, is nonlinear and dynamical, meaning the rate at a given instant depends on the preceding history of input, and not just on the instantaneous input. However, as shown by [35, 36], the firing rate of spiking neurons receiving low-pass filtered noise with fast auto-correlation timescales is approximately a function of the instantaneous input. The fast filtered noise is exactly what irregular spiking of the spiking network generates after synaptic filtering (as in Eq 15) by the fast AMPA and GABA receptors. (While our rate model does not explicitly model (irregular) spiking, it can be thought of as a mean-field approximation to a spiking network where each SSN unit or "neuron" represents a sub-population of spiking neurons, with the rate of that unit representing the average firing rate of the underlying spiking population). We thus use this static approximation to the I/O transfer function and assume the firing rates of our model units are given by Eq 2: $\boldsymbol{r}_t = F(\mathbf{h}_t^{\mathrm{total}}) = F\left(\sum_\beta \mathbf{h}_t^\beta\right)$, where $F(.)$ is the rectified power-law function of Eq 12.

We do note, however, that this static approximation can be lifted in a straightforward manner at the level of our linearized approximation (which underlies our qualitative understanding of the contrast-dependence of the gamma peak): upon linearization, a dynamic neural transfer function would result in the neural gain variables (see Eq 22) becoming frequency-dependent gain *filters*. However, as long as those gain filters are feature-less over the gamma band (*i.e.*, they vary sufficiently slowly over this band of frequencies, and in particular do not have features such as peaks within this band), their frequency dependence would not qualitatively affect the location of the gamma peak and its stimulus dependence. Thus we expect that the static I/O approximation will not alter our qualitative results.

## Modelling of gamma oscillations and local field potential

As discussed in the Introduction and Results, gamma oscillations are most consistent with noise-driven damped oscillations. We thus assumed the external input consisted of a time-independent term representing the feedforward drive due to a static stimulus, and dynamical noise:

$$
\mathbf{I}_t^\alpha = \mathbf{I}_{DC}^\alpha + \boldsymbol{\eta}_t^\alpha.
\tag{18}
$$

Given that external inputs to cortex are excitatory and only fast noise is relevant to gamma oscillations we assumed that $\boldsymbol{\eta}_t^\alpha$ was only nonzero for $\alpha = $ AMPA. We took $\boldsymbol{\eta}_t$ to have independent and identically distributed components (with zero mean) across our sub-population

units, and took it to be temporally pink noise, with fast correlation time, $\tau_{\text{corr}}$:

$$\langle \eta_i(t_1)\,\eta_j(t_2)\rangle = \delta_{ij}\sigma_\eta^2 e^{-\frac{|t_1-t_2|}{\tau_{\text{corr}}}}. \tag{19}$$

We assume that local field potential (LFP) recordings predominantly measure the inputs to the surrounding pyramidal neurons [37], and thus in our model use the current input into our excitatory units as the surrogate for LFP. More precisely we take the LFP signal at location $\mathbf{x}$ to be the total current input, $\mathbf{h}^{\text{total}}$, averaged over the $E$ neurons within a given distance of $\mathbf{x}$ (the average could be weighted with weights that decrease with distance). This can be written as the inner product of $\mathbf{h}^{\text{total}}$ with an $\mathbf{x}$-dependent weight vector:

$$LFP_t(\mathbf{x}) \propto \mathbf{e_x} \cdot \mathbf{h}^{\text{total}} = \sum_\alpha \mathbf{e_x} \cdot \mathbf{h}^\alpha, \tag{20}$$

where the weight vector $\mathbf{e_x}$ only has nonzero components for $E$ neurons that are within a given radius of location $\mathbf{x}$. In particular, in the two-population model which lacks retinotopy $\mathbf{e_x} \equiv \mathbf{e} = (1, 0)^{\text{T}}$. In the retinotopic model, we assumed that the spatial range of the LFP recording does not exceed the half-width of our the model's cortical columns (0.2 mm), and therefore we took $\mathbf{e_x}$ to be a one-hot vector with the component for the $E$ unit at location $\mathbf{x}$ equal to one, and the rest zero.

## LFP power-spectra in the linearized approximation

In order to study the power-spectra, and gain intuition about them, we linearized the dynamics around the noise-free fixed point. (Recall that we are modelling gamma as noise-driven damped oscillations, *i.e.* the network is in a regime where without noise it reaches a stable fixed point). As shown in Results, the fixed point satisfies Eq 3. The linear approximation consists of a first-order Taylor expansion in powers of the noise, $\boldsymbol{\eta}_t^\alpha$, and noise-driven fluctuations, $\delta\mathbf{h}_t^\alpha \equiv \mathbf{h}_t^\alpha - \mathbf{h}_*^\alpha$, around the stable fixed point. (Note that while the fixed point equation only involves the total current $\mathbf{h}_* \equiv \sum_\alpha \mathbf{h}_*^\alpha$, after numerically finding $\mathbf{h}_*$, we can obtain the fixed-point value of the receptor-specific currents via $\mathbf{h}_*^\alpha = W^\alpha F(\mathbf{h}_*) + I_{DC}^\alpha$). This yields

$$\tau^\alpha \frac{d\delta\mathbf{h}_t^\alpha}{dt} = -\delta\mathbf{h}^\alpha + W^\alpha \Phi \sum_\beta \delta\mathbf{h}_t^\beta + \boldsymbol{\eta}_t, \tag{21}$$

where we defined the gain matrix $\Phi$ as a diagonal matrix whose diagonal entries are

$$\Phi_{ii} \equiv F'(h_{i*}) = nk[h_{i*}]_+^{n-1} = nk^{\frac{1}{n}}r_{i*}^{1-\frac{1}{n}}. \tag{22}$$

Taking the Fourier transform of Eq 21, and solving for $\delta\tilde{\mathbf{h}}_f^\alpha$ (the Fourier transform of $\delta\mathbf{h}_t^\alpha$, where $f$ denotes frequency) we obtain

$$\delta\tilde{\mathbf{h}}_f^\alpha = \sum_\beta G^{\alpha\beta}(f)\,\tilde{\boldsymbol{\eta}}_f^\beta, \tag{23}$$

where the Green's function, $G_h^{\alpha\beta}(f)$, is given by

$$[G(f)^{-1}]^{\alpha\beta} \equiv (-i2\pi f\tau^\alpha + 1)\delta_{\alpha\beta}\mathbb{I}_{N\times N} - W^\alpha\Phi. \tag{24}$$

where $\mathbb{I}_{N\times N}$ is the identity matrix.

Since, by Eq 20, the LFP is a linear function of $\mathbf{h}^\alpha$, the power-spectrum of LFP can be written in terms of the cross-spectrum matrix of $\delta\tilde{\mathbf{h}}_f^\alpha$, which we denote by $C_h^{\alpha\beta}(f)$. Specifically

$$P_{\mathrm{LFP}}(f; \mathbf{x}) = \sum_{\alpha,\beta} \mathbf{e}_\mathbf{x}^{\mathrm{T}} C_h^{\alpha\beta}(f) \, \mathbf{e}_\mathbf{x}. \tag{25}$$

Using Eq 23 and $C_h^{\alpha\beta}(f) \propto \langle \delta\tilde{\mathbf{h}}_f^\alpha \delta\tilde{\mathbf{h}}_f^{\beta\dagger} \rangle$ we have

$$\begin{aligned} C_h^{\alpha\beta}(f) &= \sum_{\gamma,\delta} G^{\alpha\gamma}(f) \, C_\eta^{\gamma\delta}(f) \, G^{\beta\delta}(f)^\dagger, \\ &= G^{\alpha\mathrm{A}}(f) \, C_\eta^{\mathrm{AA}}(f) \, G^{\beta\mathrm{A}}(f)^\dagger. \end{aligned} \tag{26}$$

Here, $C_\eta^{\gamma\epsilon}(f)$ is the cross-spectrum of the input noise, and in the second line we relied on our assumption that noise only enters the AMPA channel, and thus only $\gamma = \delta = A \equiv \mathrm{AMPA}$ contribute to the sums. From Eq 19, we have $C_\eta^{\mathrm{A,A}}(f) = P_{\mathrm{noise}}(f) \, \mathbf{I}_{N\times N}$, where

$$P_{\mathrm{noise}}(f) = \frac{2\tau_{\mathrm{corr}} \sigma_\eta^2}{\left|1 - 2\pi i \tau_{\mathrm{corr}} f\right|^2} \tag{27}$$

is the power-spectrum of noise. We finally obtain

$$P_{\mathrm{LFP}}(f; \mathbf{x}) = P_{\mathrm{noise}}(f) \parallel \mathbf{u}_\mathbf{x}(f) \parallel^2, \tag{28}$$

where we defined

$$\mathbf{u}_\mathbf{x}(f) \equiv \sum_\beta G^{\beta\mathrm{A}}(f)^\dagger \mathbf{e}_\mathbf{x}. \tag{29}$$

## Definition of suppression index, and gamma peak frequency and width

Suppression index was based on the size tuning curve, $r(R)$, of the center $E$ or $I$ units, measured for gratings of 100% contrast. It was defined by

$$\mathrm{SI} = 1 - \frac{r(R_{\max})}{\max_R r(R)} \tag{30}$$

where $R$ is the grating radius, $r(R)$ is the size tuning curve, and $R_{\max}$ is the maximum grating radius used.

As in [1], we identified the gamma peak frequency with the frequency (within the extended gamma band 10–100 Hz) at which the difference of the evoked ($c > 0$) and spontaneous ($c = 0$) LFP log-spectra (or the ratio of those spectra) is maximized:

$$f_{\mathrm{peak}}(c) = \arg \max_f [\log P_{\mathrm{LFP}}(f; c) - \log P_{\mathrm{LFP}}(f; c = 0)]. \tag{31}$$

As a measure of gamma peak width (or half-width) at contrast $c$, we used the half-width at half-height of the relative power spectrum $\frac{P_{\mathrm{LFP}}(f;c)}{P_{\mathrm{LFP}}(f;0)}$.

## The eigen-decomposition of the LFP power-spectrum

The linearized dynamics of Eq 4 can be written in terms of the Jacobian matrix, $\mathcal{J}$ as

$$\frac{d\delta\mathbf{h}_t^\alpha}{dt} = \sum_\beta \mathcal{J}_{\alpha\beta} \, \delta\mathbf{h}_t^\beta \tag{32}$$

where

$$\mathcal{J}_{\alpha\beta} \equiv \tau_\alpha^{-1}(-\delta_{\alpha\beta}\mathbb{I}_{N\times N} + W^\alpha\Phi). \tag{33}$$

(with $\alpha, \beta \in \{\text{AMPA, GABA, NMDA}\}$) is the $(\alpha, \beta)$ block (an $N \times N$ matrix) of the full $3N \times 3N$ Jacobian matrix. The normal modes of this linear system correspond to eigenvectors of the Jacobian, which evolve in time according to $e^{\lambda t}$ which for $\lambda = \gamma + i\omega_0$ with real $\gamma$ and $\omega_0$ can be written $e^{-|\gamma|t}(\cos \omega t + i \sin \omega t)$ (we used the fact that stability require that the eigenvalue real part, $\gamma$, is negative). We thus see that this mode oscillates at (angular) frequency $\omega_0 = \text{Im}\lambda$, and decays at the rate given by $|\gamma| = -\text{Re}\lambda$. (For brevity, in this section we will use the angular frequency $\omega = 2\pi f$ instead of $f$). Comparison of Eqs 4 and 24 shows that the inverse Green's function can be written in terms of the Jacobian as $[G(f)^{-1}]_{\alpha\beta} = \tau_\alpha(-i\omega\mathbb{I} - \mathcal{J})_{\alpha\beta}$. Defining a diagonal matrix $T$ with the first, second, and last third of its diagonal elements given by $\tau_{\text{AMPA}}$, $\tau_{\text{GABA}}$ and $\tau_{\text{NMDA}}$, respectively. We can write

$$G(f) = (-i\omega\mathbb{I} - \mathcal{J})^{-1}T^{-1} \tag{34}$$

We start by rewriting the Green's function in terms of the eigen-decomposition of the Jacobian $\mathcal{J} = V\Lambda V^{-1}$, where $\Lambda$ is the diagonal matrix of eigenvalues, $\lambda_a$, and $V$ is a matrix with columns given by corresponding (right) eigenvectors. Equivalently we can write $\mathcal{J} = \sum_a \lambda_a \mathbf{R}_a \mathbf{L}_a$, where the right eigenvectors $\mathbf{R}_a$ are the columns of $V$ and the left eigenvectors, $\mathbf{L}_a$, are the rows of $V^{-1}$. Using this decomposition we obtain

$$G(f) \quad = \sum_a \frac{1}{-i\omega - \lambda_a}\mathbf{R}_a\mathbf{L}_a T^{-1}. \tag{35}$$

We can then rewrite Eq 29 as

$$u_\mathbf{x}(f)^\dagger = \tau_{\text{AMPA}}^{-1}\sum_a \frac{\mathcal{R}_a(\mathbf{x})}{-i\omega - \lambda_a}\mathbf{L}_a^A \tag{36}$$

where $\mathbf{L}_a^A$ is the row-vector formed by the AMPA components of $\mathbf{L}_a$, and we defined

$$\mathcal{R}_a(\mathbf{x}) = \sum_\beta [\mathbf{R}_a]_{\beta,E,\mathbf{x}} \tag{37}$$

namely, the (possibly complex) scalar function $\mathcal{R}_a(\mathbf{x})$ is the $E$, $\mathbf{x}$ component (component on $E$ subpopulation at column $\mathbf{x}$) of the right-eigenvector after summing over receptor indices. Here we have assumed that the LFP probe is completely local and reflects total current (hence the sum over $\beta$) into $E$ (pyramidal) neurons of the recorded column $\mathbf{x}$. The limitation to AMPA components of the left eigenvectors, on the other hand, reflects our assumption that external noise is entering only via AMPA receptors. Substituting in Eq 28, we then obtain

$$\frac{P_{\text{LFP}}(f; \mathbf{x})}{P_{\text{noise}}(f)} = \| \mathbf{u}_\mathbf{x}(f) \|^2 \tag{38}$$

$$= \sum_{a,b} A_{ab}(x)\frac{1}{i\omega - \lambda_b^*}\frac{1}{-i\omega - \lambda_a} \tag{39}$$

where we defined

$$A_{ab}(x) = \tau_{\text{AMPA}}^{-2}\langle L_b^A, L_a^A \rangle \ \mathcal{R}_b^*(\mathbf{x})\mathcal{R}_a(\mathbf{x}) \tag{40}$$

and $\langle L_b^A, L_a^A \rangle = L_a^A (L_b^A)^\dagger$ is the Hermitian inner product of the two left eigenvectors, within the AMPA subspace. The key factor here is $\mathcal{R}_b^*(\mathbf{x})\mathcal{R}_a(\mathbf{x})$ which determines the $\mathbf{x}$-dependence and can affect the local dependence of power spectrum on information at point $\mathbf{x}$.

Eqs 39 and 40 constitute our main result here. They express the ratio of LFP to noise power spectrum as a sum of contributions by pairs of eigen-modes, and contributions by individual eigen-modes (the latter corresponding to the diagonal summands with $a = b$).

In particular, the individual contribution of mode $a$ is

$$P_a(f; \mathbf{x}) \equiv \frac{A_{aa}(\mathbf{x})}{|-i\omega - \lambda_a|^2} = \frac{A_{aa}(\mathbf{x})}{\gamma_a^2 + (\omega - \omega_a)^2}, \tag{41}$$

where $\gamma_a$ and $\omega_a$ are minus the real and imaginary components of $\lambda_a$, and $f = \frac{\omega}{2\pi}$. This contribution is a Lorentzian function that peak at the natural frequency frequency $\omega_a$ and has half-width $\gamma_a$. The amplitude of this peak is given by

$$\frac{A_{aa}(\mathbf{x})}{\gamma_a^2} \propto \frac{|\mathcal{R}_a(\mathbf{x})|^2}{\gamma_a^2}, \tag{42}$$

where we used Eq 40. This amplitude is thus proportional to $|\mathcal{R}_a(\mathbf{x})|^2$, *i.e.*, the squared absolute value of the corresponding right eigenvector component at location $\mathbf{x}$. It is also inversely proportional to the squared half-width which measures the distance between the eigenvalue and the imaginary axis. Thus eigenvalues closer to this axis produce stronger and sharper peaks, which appear in the LFP spectrum probed at location $\mathbf{x}$ if the corresponding right eigenvector has strong components at that location.

The sum in Eq 39 also contains terms each of which can be interpreted as the contribution of a pair of (distinct) modes. When the left-eigenvectors of different modes are orthogonal (according to the inner product defined after Eq 40, which corresponds to the orthogonality of the AMPA components of the vectors under the common inner product) these contributions vanish. More generally, the contribution of the pair $(a, b)$ can be written as (making use of $A_{ba}(\mathbf{x}) = A_{ab}(\mathbf{x})^*$)

$$P_{ab}(f; \mathbf{x}) \equiv 2\text{Re}\, A_{ab}(\mathbf{x}) \frac{1}{i\omega - \lambda_b^*} \frac{1}{-i\omega - \lambda_a} \tag{43}$$

$$= 2 \frac{N_{ab}^R(f; \mathbf{x}) + N_{ab}^I(f; \mathbf{x})}{D_{ab}(f)}, \tag{44}$$

where we defined

$$D_{ab}(f) = (\gamma_a^2 + (\omega - \omega_a)^2)(\gamma_b^2 + (\omega - \omega_b)^2), \tag{45}$$

and

$$N_{ab}^R(f; \mathbf{x}) = \text{Re}[A_{ab}(\mathbf{x})](\gamma_a \gamma_b + (\omega - \omega_a)(\omega - \omega_b)), \tag{46}$$

$$N_{ab}^I(f; \mathbf{x}) = \text{Im}[A_{ab}(\mathbf{x})](\gamma_a(\omega - \omega_b) - \gamma_b(\omega - \omega_a)). \tag{47}$$

Alternatively, the pair contribution $P_{ab}(f; \mathbf{x})$ is given by the product of the individual contributions $P_a(f; \mathbf{x})$ and $P_b(f; \mathbf{x})$, with a correction factor given by $2A_{aa}^{-1}(\mathbf{x})A_{bb}^{-1}(\mathbf{x})(N_{ab}^R(f; \mathbf{x}) + N_{ab}^I(f; \mathbf{x}))$.

## Parametrization of the two-population and retinotopic models

The 2x2 (full) connectivity matrix of the 2-population model is parametrized by the four parameters $J_{ab}$ ($a, b \in \{E, I\}$) as follows:

$$W = \begin{pmatrix} J_{EE} & -J_{EI} \\ J_{IE} & -J_{II} \end{pmatrix}. \tag{48}$$

The DC stimulus input corresponds to feedforward excitatory inputs from LGN and targets both sub-populations only via the AMPA channel (since this input is time-independent, its distribution across NMDA and AMPA channels is actually of no consequence). As in the original SSN, we assumed this input scales linearly with contrast, $c$, but with varying relative strengths to the $E$ and $I$ captured by the two parameters $g_E$ and $g_I$:

$$\mathbf{I}_{DC} = c \begin{pmatrix} g_E \\ g_I \end{pmatrix}. \tag{49}$$

In the retinotopic model we index the neurons by their $E/I$ type and retinotopic location. We parametrized the recurrent connection weight from the pre-synaptic $E$ and $I$ units at location $\mathbf{y}$ to the type $a$ ($a \in [E, I]$) post-synaptic unit at location $\mathbf{x}$ by

$$W_{\mathbf{x},a|\mathbf{y},E} \propto J_{a,E} \left[ \lambda_{a,E}\, \delta_{\mathbf{x},\mathbf{y}} + (1 - \lambda_{a,E}) e^{-\frac{\|\mathbf{x}-\mathbf{y}\|}{\sigma_{a,E}}} \right] \tag{50}$$

for excitatory projections, and

$$W_{\mathbf{x},a|\mathbf{y},I} \propto J_{a,I}\, e^{\frac{(\mathbf{x}-\mathbf{y})^2}{2\sigma_{a,I}^2}}, \tag{51}$$

for inhibitory ones. We are using proportionality instead of equal signs in the above equations, because a normalization was done such that the total weight of each type received by a unit was given by the corresponding $J_{ab}$ (independent of the $\sigma_{ab}$ and $\lambda_{ab}$ parameters). Recurrent connectivity was thus parametrized by the 2x2 matrices $J_{ab}$ and $\sigma_{ab}$, the two $\lambda_{a,E}$, and the NMDA fraction, $\rho_N$, 11 parameters in total. For $\sigma_{II}$ and $\sigma_{EI}$ we used values (see S1 Table) small compared to the distance between neighboring columns (0.4 mm) so that inhibition was effectively local (*i.e.*, intra-columnar); we did not vary $\sigma_{II}$ and $\sigma_{EI}$ across our randomly sampled networks.

We modeled the external stimulus input to the type $a$ unit at $\mathbf{x}$ by

$$I_{a,\mathbf{x}}^{DC} = c\, g_a\, I_{\mathbf{x}} \qquad (a \in \{E, I\}). \tag{52}$$

where $g_E$ and $g_I$ reflect the relative strengths of feedforward connections received by V1's excitatory and inhibitory networks, and $I_{\mathbf{x}}$ captures the spatial profile of the visual stimulus. For a grating of radius $r_{\mathrm{grat}}$ we modeled the spatial contrast profile, $I_{\mathbf{x}}$, as

$$I_{\mathbf{x}} = \frac{1}{1 + e^{\frac{\|\mathbf{x}\| - r_{\mathrm{grat}}}{w_{\mathrm{RF}}}}}. \tag{53}$$

The parameter $w_{\mathrm{RF}}$ smooths the edges of the grating (due to feedforward filtering by receptive fields of width $\sim w_{\mathrm{RF}}$). Note that for $r_{\mathrm{grat}} \gg w_{\mathrm{RF}}$ (which was true for most grating sizes employed), local contrast is nearly uniform under the support of the external input (except

within a "boundary layer" of width $w_{RF}$ near $\|x\| = r_{grat}$). For the Gabor stimulus we have

$$I_{\mathbf{x}} = e^{-\frac{\|\mathbf{x}\|^2}{2\sigma_{Gabor}^2}}. \tag{54}$$

We took $\sigma_{Gabor} = 0.5°$, as in [1], and the peak contrast ($c$ in Eq 52) was always 100% for this stimulus.

## Model parameters and parameter sampling

see S1 Table below for the values of all parameters or parameter ranges for models used in different figures. For the models used in Figs 1 and 4, we found their parameters ($J_{ab}$ and $g_a$ which are shared in both figures, and $\sigma_{a,E}$ and $\lambda_{a,E}$ for Fig 4) using random sampling (as further described below) searching for networks that would exhibit the local contrast-dependence of the gamma peak together with strong surround suppression.

For studying the robustness of the contrast dependence of gamma frequency in the two-population model in Fig 2, and in the case of the retinotopic SSN with a smooth fall-off of excitatory horizontal connectivity in Fig 6, we sampled parameters from wide biologically plausible ranges. To determine these ranges for the recurrent and feedforward weights, we first made rough biological estimates for the recurrent $E$ and $I$ weights (*i.e.*, $J_{aE}$ and $J_{aI}$, respectively, for $a \in \{E, I\}$), as well as the (excitatory) feedforward weights ($g_E$ and $g_I$); we denote these estimates by $J_E^*$, $J_I^*$ and $g^*$, respectively. We then independently varied parameters controlling each type of weight between 0.5 to 1.5 times those estimates (see S1 Table for the actual values).

To come up with the mid-range estimates, $J_E^*$, $J_I^*$ and $g^*$, we relied on empirical estimates of the effect of recurrent and feedforward inputs on the membrane voltage of a post-synaptic neuron. Note that while $J_{ab}$ (and thus $J_E^*$ and $J_I^*$), have dimensions of voltage (such that the recurrent input $W\,\boldsymbol{r}$ has our units of current), $g_a$ (and thus $g^*$) have dimensions of current. In our model, we measured the currents in units of mV/s, by including an implicit factor of membrane capacitance in them. The membrane potential response to a unit current is normally given by the membrane resistance, which in our units becomes the membrane time constant, which we take to be $\tau_m = 0.01$ s. So to obtain an estimate of $g^*$ from voltage measurements, we need to divide the estimate by $\tau_m = 0.01$ s.

We estimated the effect of feedforward inputs on membrane voltage using measurements in cats and mice [41, 51–54] (see [55] for a review and discussion of these measurements). Based on these measurements, we estimate the maximum feedforward input, achieved for 100% contrast to be on the order of the rest to threshold distance, which is around 20 mV [56]. This yields $g^* \sim \tau_m^{-1} 20\,\text{mV}/(100\%) = 20$ mV/s per 1% contrast. The $J_{ab}$ parameters measure the total synaptic weight, which biologically is given by a unitary excitatory or inhibitory (depending on $b$) post-synaptic potential (EPSP or IPSP) times the total number of pre-synaptic V1 neurons, $K_b$, of type $b$. Based on anatomical measurements for sensory cortex (reviewed in [55]) we estimate the effective $K_E$ to be $\sim 400$ (with a wide margin of uncertainty). And based on electrophysiological measurements we assume the median EPSP amplitude to be $\sim 0.5$ mV. This yields $J_E^* = 0.5 \times 400 = 200$ mV. For unitary IPSP amplitude, we used the same value of 0.5 mV, but assumed half as many inhibitory pre-synaptic inputs, due to the smaller number of inhibitory cells in the circuit. We thus took $J_I^* = J_E^*/2$.

In Fig 5, the four extra parameters controlling the spatial profile of excitatory horizontal connections in the retinotopic SSN were additionally sampled randomly as well. These parameters are $\lambda_{EE}$ and $\lambda_{IE}$ quantifying the intra-columnar excess connectivity, and $\sigma_{IE}$ and $\sigma_{EE}$ quantifying the length-scale (range) of the long-range components of excitatory recurrent connections. We sampled the two $\sigma$'s uniformly between 150 $\mu$m and 600 $\mu$m. The non-

columnar model had $\lambda_{EE} = \lambda_{IE} = 0$, while for the columnar model we sampled these uniformly from the interval [0.25, 0.75].

Finally, we assumed that recurrent V1 excitatory synapses are dominated by AMPA, rather than NMDA, and therefore sampled $\rho_N$ uniformly at random in the interval [0.3, 0.5].

All parameters were sampled uniformly and independently over their ranges, except for enforcement (by sample rejection) of three inequality constraints:

$$J_{EI}J_{IE} > J_{EE}J_{IE},$$
$$J_{II}g_E > J_{EI}g_I,$$
$$\sigma_{IE} > \sigma_{EE}.$$

Previous work has shown that the first inequality promotes stability (almost a necessary condition) [23, 57], the second inequality ensures that the network is not too strongly inhibition-dominated such that excitatory rates become too small [23, 57]. The last inequality is necessary for obtaining considerable surround suppression [22].

## Eigenvalue spectra of rate-based and multi-receptor SSNs in the presence and absence of NMDA

Here we prove that the spectrum of a linearized synaptic model without NMDA is the same as the spectrum of a linearized $E/I$ rate model, with the exchange $\tau_{AMPA} \rightarrow \tau_E$ and $\tau_{GABA} \rightarrow \tau_I$. This means that, in particular, the formulae of [16] for eigenvalues in a 2-neuron/population model still hold for this model with the above replacements.

We start by rewriting the inverse Green's function, using the Green's function defined in Eq 24, which we now write in full matrix form. We will also start general, allowing for $q$ different receptor types (we also write in terms of the angular frequency $\omega = 2\pi f$).

$$G(\omega)^{-1} = A - \mathbf{W}\Phi P \tag{55}$$

where we define

$$A := -i\omega\mathbf{T} + \mathbb{I} \qquad \in \mathbb{R}^{qN \times qN} \tag{56}$$

$$\mathbf{W} := \begin{pmatrix} W^A \\ W^G \\ W^N \\ \vdots \end{pmatrix} \qquad \in \mathbb{R}^{qN \times N} \tag{57}$$

$$P := I_{N \times N} \otimes \mathbf{1}_q^T = (\mathbb{I}_{N \times N}, \mathbb{I}_{N \times N}, \mathbb{I}_{N \times N}, \ldots) \qquad \in \mathbb{R}^{N \times qN} \tag{58}$$

where $\mathbf{T} = \text{diag}(\boldsymbol{\tau}_s) \otimes I_{N \times N}$ and $\boldsymbol{\tau}_s \in \mathbb{R}$ (in our case $\boldsymbol{\tau}_s = (\tau_A, \tau_G, \tau_N)$ or $(\tau_A, \tau_G)$), and $\mathbf{1}_q^T = (1, \ldots, 1) \in \mathbb{R}^q$.

The eigenvalue spectrum correspond to values of $z = -i\omega$ which make the determinant of $G(\omega)^{-1}$ vanish. Noting that the second term in Eq 55 is rank-deficient (has at most rank $N$, instead of full-rank $qN$), we make use of the "matrix determinant lemma" to write:

$$\det(G^{-1}(\omega)) = \det(A)\det(\mathbb{I}_{N \times N} - PA^{-1}\mathbf{W}\Phi) \tag{59}$$

It is not hard to see that

$$PA^{-1} = \left( \frac{\mathbb{I}_{N \times N}}{-i\omega\tau_A + 1}, \frac{\mathbb{I}_{N \times N}}{-i\omega\tau_G + 1}, \frac{\mathbb{I}_{N \times N}}{-i\omega\tau_N + 1}, \cdots \right) \tag{60}$$

and therefore

$$PA^{-1}\mathbf{W} = \sum_{\alpha=1}^{q} \frac{1}{-i\omega\tau_\alpha + 1} W^\alpha \tag{61}$$

We now limit to $q = 2$ with only AMPA and GABA.

$$PA^{-1}\mathbf{W} = \frac{1}{-i\omega\tau_A + 1} W^A + \frac{1}{-i\omega\tau_G + 1} W^G = W\widetilde{A}^{-1} \tag{62}$$

where we have made use of the specific forms of $W^A$ and $W^G$ (namely, that they have zero columns for inhibitory and excitatory neurons, respectively) from Eq 17, and where we have defined

$$W = \sum_\alpha W^\alpha \qquad \in \mathbb{R}^{N \times N} \tag{63}$$

$$\widetilde{A} := z\tilde{T} + \mathbb{I}_{N \times N} \tag{64}$$

with $\tilde{T} = \mathrm{diag}(\tilde{\tau}) \in \mathbb{R}^{N \times N}$ where $\tilde{\boldsymbol{\tau}} = (\tau_A, \dots, \tau_G, \dots) \in \mathbb{R}^N$ is the $N$-dimensional vector with first $N_E$ components equal to $\tau_A$ and the last $N_I$ components equal to $\tau_G$. After identifying $\tau_{A/G}$ with $\tau_{E/I}$, we thus see that $\tilde{T}$ is the same as the $T$ matrix of the $r$-model (which is $N$-dimensional), as is $W$ its connectivity matrix. Also noting that $(z\tilde{T} + \mathbb{I}_{N \times N})^{-1}$ and $\Phi$ are both diagonal, we can commute them in Eq 62 to obtain:

$$\det(G^{-1}(\omega)) = \det(A)\det(\mathbb{I}_{N \times N} - W\Phi\widetilde{A}^{-1}) \tag{65}$$

$$= \frac{\det(A)}{\det(\widetilde{A})} \det(z\widetilde{T} + \mathbb{I}_{N \times N} - W\Phi) \tag{66}$$

$$= \frac{\det(A)}{\det(\widetilde{A})} \det(z\widetilde{T} + \mathbb{I}_{N \times N} - \Phi W) \tag{67}$$

(to get the last line, do a similarity transform with $\Phi$, of the matrix in the last determinant).

Now it is explicit that the zeros of the last determinant factor are the eigenvalues of the $N$-dimensional $r$-system (after $\tau_{A/G} \leftrightarrow \tau_{E/I}$ identification).

The first factor, on the other hand, can be written as:

$$\frac{\det(A)}{\det(\tilde{A})} = \frac{(z\tau_A + 1)^N (z\tau_G + 1)^N}{(z\tau_A + 1)^{N_E} (z\tau_G + 1)^{N_I}} = (z\tau_A + 1)^{N_I} (z\tau_G + 1)^{N_E} \tag{68}$$

So the spectrum also has $N$ additional *real* eigenvalues (in addition to those of the $r$-model) with values $-\tau_A^{-1}$ and $-\tau_G^{-1}$, and multiplicities, $N_I$ and $N_E$, respectively. (Thus in total we have $2N$ eigenvalues as we should).

In particular, all oscillatory/complex eigenvalues are exactly those of the $r$-model in the no-NMDA case, which in the 2-neuron case are given by the formulae in [16].

When the model has NMDA (or other) receptors (*i.e.*, for $q > 2$), the above exact correspondence will break down. However, as we now show, the relative slowness of NMDA allows for approximate reductions to the case of two receptor model in different frequency regimes. We consider two regimes for the effect of NMDA:

1. when $|z|$ or $\omega$ are very small compared to the NMDA time-constant: $\omega \ll \tau_N^{-1}$.

2. when $|z|$ or $\omega$ are very large compared to the NMDA time-constant: $\omega \gg \tau_N^{-1}$.

The first regime is relevant for DC response and DC properties (such as surround suppression of steady-state rates). The second regime is approximately valid for gamma oscillations, thanks to the relatively high frequency of those.

In regime 1, it is obvious that the breakdown of $E$ weights into the two types does not have any effects, simply because (setting $\omega$ to 0) time-scales do not play any role here. So the parameter $\rho_N$ makes no difference to fixed point response properties.

In regime 2, looking at Eq 61, we note that the prefactor $\frac{1}{-i\omega\tau_\alpha+1}$ for NMDA is very small and can be ignored. This means that for high frequencies (*e.g.*, approximately frequencies around gamma) we can simply kill all NMDA weights, and only consider the AMPA weight matrix, $W^A$. In particular, the model where $W^A \propto W^N$, then the effect of NMDA on the gamma peak is approximately equivalent to reducing total excitatory weights (which all affect DC properties) by a scalar factor (which in our formalism is $1 - \rho_N$) when it comes to gamma properties.

## Analysis of gamma peak frequency in the two-population model

We consider now the case of the two-population model presented in Two-population model. We will also assume no NMDA contribution (or equivalently work in the very slow NMDA regime, and replace all excitatory weights with their AMPA part, as explained at the end of the previous subsection).

In this case the gamma peak frequency is closely approximated by the imaginary part of the eigenvalues of the Jacobian matrix:

$$\mathcal{J} = -\mathbf{T}^{-1} + \mathbf{T}^{-1}\mathbf{W}\Phi \tag{69}$$

$$= \begin{pmatrix} \gamma_E(-1 + W_{EE}\Phi_E) & -\gamma_E W_{EI}\Phi_I \\ \gamma_I W_{IE}\Phi_E & \gamma_I(-1 - W_{II}\Phi_I) \end{pmatrix} \tag{70}$$

where we defined $\gamma_E \equiv \tau_{AMPA}^{-1}$ and $\gamma_I \equiv \tau_{GABA}^{-1}$. Noting that the trace and determinant of $\mathcal{J}$ yield the sum and product of the eigenvalues, respectively, we obtain the expression (see [16])

$$2\lambda_{1,2} = \gamma_E(W_{EE}\Phi_E - 1) - \gamma_I(W_{II}\Phi_I + 1)$$
$$\pm\sqrt{[\gamma_E(W_{EE}\Phi_E - 1) + \gamma_I(W_{II}\Phi_I + 1)]^2 - 4\gamma_E\gamma_I W_{EI}W_{IE}\Phi_E\Phi_I} \tag{71}$$

A gamma peak exists only if the expression under the square root is negative, *i.e.*

$$4\gamma_E\gamma_I W_{EI}W_{IE}\Phi_E\Phi_I > [\gamma_E(W_{EE}\Phi_E - 1) + \gamma_I(W_{II}\Phi_I + 1)]^2, \tag{72}$$

in which case, for the gamma peak angular frequency $\omega_0$, we (approximately) have

$$4\omega_0^2 = 4\beta_E\beta_I W_{EI}W_{IE} - (\beta_E W_{EE} + \beta_I W_{II} + \gamma_I - \gamma_E)^2 \tag{73}$$

where we defined $\beta_X := \gamma_X\Phi_X$ for $X \in \{E, I\}$.

We will now obtain a simplified expression for the derivative of $\omega_0^2$ with respect to the contrast $c$, using the rectified supralinear nonlinearity of the SSN. Using $\Phi_* = nk^{\frac{1}{n}}r_*^{1-\frac{1}{n}}$ (where $r_*$ is the firing rate at fixed point) we obtain

$$\frac{d\beta_*}{dc} = \frac{n-1}{n}\beta_*\frac{d\ln r_*}{dc} \tag{74}$$

Then using Eq 73, and defining

$$A := (\beta_E W_{EE} + \beta_I W_{II} + \gamma_I - \gamma_E) \tag{75}$$

and $(\dots)' := \frac{d(\dots)}{dc}$, we find:

$$\frac{n}{n-1}\frac{d\omega_0^2}{dc} = \beta_E\beta_I W_{EI}W_{IE}(\ln r_E + \ln r_I)'$$
$$- \frac{2A}{4}\left(\beta_E W_{EE}(\ln r_E)' + \beta_I W_{II}(\ln r_I)'\right) \tag{76}$$

$$= \omega_0^2\ (\ln r_E + \ln r_I)'$$
$$+ \frac{1}{2}A^2\left[\frac{(\ln r_E)' + (\ln r_I)'}{2} - \frac{\sum_a w_a(\ln r_a)'}{\gamma_I - \gamma_E + \sum_a w_a}\right] \tag{77}$$

where the sums are over $a \in \{E, I\}$ and we defined

$$w_a := \beta_a W_{aa} \qquad a \in \{E, I\} \tag{78}$$

Let us now focus on the sign of $\frac{d\omega_0^2}{dc}$. Assuming that we are in the gamma oscillatory regime (*i.e.*, $\omega_0$ is real) and that the fixed point rates increase with contrast, then from Eq 77 we find that sufficient conditoin for $\frac{d\omega_0^2}{dc} > 0$ is that the factor in the square brackets in Eq 77 is positive. In the solutions of SSN most relevant to cortical biology, $(\ln r_I)'$ tends to be larger than $(\ln r_E)'$ (because excitatory rates tend to saturate or supersaturate earlier). We thus consider two extreme cases: $(\ln r_E)' = (\ln r_I)'$ and $(\ln r_E)' = 0$.

In the first case, the bracket becomes $(\ln r_I)'\left[1 - \frac{\sum_a w_a}{\gamma_I - \gamma_E + \sum_a w_a}\right] = (\ln r_I)'\frac{\gamma_I - \gamma_E}{\gamma_I - \gamma_E + \sum_a w_a}$, which is positive as long as $\gamma_I > \gamma_E$ (which is unfortunately not the case for GABA and AMPA).

In the second case, the bracket factor becomes $(\ln r_I)'\frac{\gamma_I - \gamma_E + \sum_a w_a - 2w_I}{2(\gamma_I - \gamma_E + \sum_a w_a)} = (\ln r_I)'\frac{\gamma_I - \gamma_E + w_E - w_I}{2(\gamma_I - \gamma_E + \sum_a w_a)}$. This is positive (as long as the denominator is positive, which is true as long as $\gamma_I > \gamma_E$) if

$$w_E - \gamma_E + \gamma_I > w_I \tag{79}$$

But the stability of the fixed point dictates that the expression on first line of Eq 71 (the real part of the eigenvalues) has to be negative and thus

$$w_I > w_E - \gamma_E - \gamma_I. \tag{80}$$

## Supporting information

**S1 Table. Parameters of models used in different figures of the main text.** In Figs 2, 3 and 5, parameters were sampled independently and uniformly from the ranges given in the table, except for enforcing three inequality constraints (*i.e.*, sampled parameter sets violating any of

these inequalities were rejected). See the main text (Methods) for details. †: these were the ranges for sampled $\lambda_{EE}$ and $\lambda_{IE}$ of the columnar model; these parameters were zero in the non-columnar model.
(PDF)

**S1 Fig. Behaviour of retinotopic V1 models with long-range inhibitory connections.** The format of the figure is exactly the same as in Fig 5 of the main text (and the reader is referred to the caption of that figure for the detailed guide). Similar to that main figure, this figure compares the locality of gamma contrast dependence in models with and without boosted intra-columnar recurrent excitatory connectivity (columnar vs. non-columnar models, respectively) across their parameter space. However, unlike those in the main figure, the sampled models here had long-range inhibitory connections. Specifically, in each sample the range of $I \rightarrow E$ and $I \rightarrow I$ connections were set to two-thirds of the randomly-sampled ranges of the $E \rightarrow E$ and $E \rightarrow I$ excitatory connections, respectively (as in the main text, the excitatory ranges, alongside other parameters, were sampled randomly over a broad range). Thus, while $I$ connections were 33% shorter than $E$ connections, they had long and variable ranges across samples; by contrasts, in the main Fig 5, both $I \rightarrow E$ and $I \rightarrow I$ connections had a constant range of 0.09 mm (c.f. our mini-column size of 0.4 mm), across all sampled model. As evident, *e.g.*, from the stark contrast between the behaviour of samples in panels E vs. J (compare with the same panels in Fig 5 of main text), the qualitative difference between the columnar vs. non-columnar models (in accounting for the local contrast dependence of gamma frequency) remains unchanged in the presence of long-range inhibition; our conclusions are thus robust with respect to the assumption of very short inhibitory connections.
(PDF)

**S2 Fig. Hopf bifurcation diagram.** The left and right columns show the Hopf bifurcation diagrams for the two example models used in Figs 4 and 6 of the main text, respectively, when stimulated with gratings of full contrast. As the strength of recurrent excitation, $J_{EE}$, is increased beyond a crticial value, the network undergoes a Hopf bifurcation, switching from a state of damped oscillations to a state with sustained oscillations. To obtain these plots, at each value of $J_{EE}$, the networks were simulated without noise for long enough to reach steady state: below the Hopf bifurcation the steady state corresponds to a stable fixed point, while above the bifurcation it corresponds to a stable limit cycle (2 seconds of network dynamics were simulated, containing many tens of oscillation cycles). In the plots of the top row the lower and upper branches of the red lines show the minimum and maximum excitatory firing rates (of the $E$ unit in the center of the model's retinotopic grid) throughout the steady state oscillations (below the bifurcation these lines overlap, as the oscillation amplitude is zero and maximum and minimum rates are equal). In the bottom row plots, the firing rate is replaced with the LFP signal recorded at the center of the grid. The values on the x axes show the factor by which $J_{EE}$ was amplified over the original values in Figs 4 and 6 of the main text. The vertical solid blue lines in the left and right columns correspond to the values of $J_{EE}$ used in the S3 and S4 Figs, respectively (the dashed lines correspond to the original values used in the main figures).
(PDF)

**S3 Fig. Behaviour of noise-driven oscillations the model of Fig 4 (main text) when pushed above the Hopf bifurcation.** Except for the bottom right plot, the format of the rest of the figure is the same as in panels B-D and F-G of Fig 4 of the main text (and the reader is referred to the caption of that figure for the detailed guide). The parameters of the model are also the same as those in Fig 4, except for $J_{EE}$ which has been strengthened by a relative factor of 1.112. Unlike in Fig 4, the full stochastic dynamics of the model network were simulated (for 60

seconds, in order to allow for accurate estimation of power-spectra). In particular, the LFP power-spectra shown in the the left column plots were obtained from these simulations using the Welch periodogram method, followed by a gaussian smoothing with a $\sigma$ of 5 Hz (the dashed lines show the Welch periodogram without smoothing; conclusions are not sensitive to this smoothing). The peak frequencies were obtained using the methods described in the main text from these power-spectra. The bottom-right plot shows a portion of the simulated trajectory in the plane of the $E$ and $I$ firing rates in the center of the retinotopic grid. These show many cycles of the noise-driven oscillations (without input noise, the same plots would have shown overlapping deterministic trajectories going around a diagonally elongated oval-shaped limit cycle).
(PDF)

**S4 Fig. Behaviour of the model of Fig 6 (main text) when pushed above the Hopf bifurcation.** Except for the bottom-right plot, the format of the rest of the figure is the same as in Fig 6 of the main text (and the reader is referred to the caption of that figure for the detailed guide). The parameters of the model are also the same as those in Fig 6, except for $J_{EE}$ which has been strengthened by a relative factor of 1.037. The simulation procedure and the description of the bottom-right plot are as given in the caption of S3.
(PDF)

## Acknowledgments

We thank Takafumi Arakaki for technical help throughout this research, Luca Mazzucato for valuable comments on the manuscript, and Guillaume Hennequin for generously sharing LATEX code. CH acknowledges technical help and invaluable feedback from Gabriel Barello, Elliott Abe, and David Wyrick.

## Author Contributions

**Conceptualization:** Kenneth D. Miller, Yashar Ahmadian.

**Data curation:** Caleb J. Holt, Yashar Ahmadian.

**Formal analysis:** Yashar Ahmadian.

**Funding acquisition:** Kenneth D. Miller, Yashar Ahmadian.

**Investigation:** Caleb J. Holt, Kenneth D. Miller, Yashar Ahmadian.

**Methodology:** Kenneth D. Miller, Yashar Ahmadian.

**Project administration:** Yashar Ahmadian.

**Software:** Caleb J. Holt, Yashar Ahmadian.

**Supervision:** Yashar Ahmadian.

**Visualization:** Caleb J. Holt, Yashar Ahmadian.

**Writing – original draft:** Caleb J. Holt, Kenneth D. Miller, Yashar Ahmadian.

**Writing – review & editing:** Kenneth D. Miller, Yashar Ahmadian.

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
