## [Decision Letter · Decision Letter 0]

28 Nov 2023

Dear Dr. Ahmadian,

Thank you very much for submitting your manuscript "The stabilized supralinear network accounts for the contrast dependence of visual cortical gamma oscillations" for consideration at PLOS Computational Biology.

As with all papers reviewed by the journal, your manuscript was reviewed by members of the editorial board and by several independent reviewers. In light of the reviews (below this email), we would like to invite the resubmission of a significantly-revised version that takes into account the reviewers' comments.

We cannot make any decision about publication until we have seen the revised manuscript and your response to the reviewers' comments. Your revised manuscript is also likely to be sent to reviewers for further evaluation.

Sincerely,

Tatiana Engel

Guest Editor

PLOS Computational Biology

Thomas Serre

Section Editor

PLOS Computational Biology

Reviewer's Responses to Questions

**Comments to the Authors:**

Reviewer #1: The paper provides a novel mechanistic explanation for the relationship between visual stimulus contrast and gamma oscillation frequency in neural activity recorded in macaque primary visual cortex. A stabilized supralinear network (SSN) robustly captures the contrast dependence of gamma peak frequency, and a retinotopically organized SSN model with both short-range and long-range excitatory horizontal connections can exhibit both surround suppression and the local contrast dependence of gamma peak frequency. The paper is clearly written. The use of linearized equations to draw insights into the SSN's behavior aids understanding, providing further insights into the proposed cortical circuit mechanisms. The work has the potential to bridge the gap between empirical neurophysiological observations and computational models in the field of visual neuroscience.

Questions.

Could the authors elaborate on whether their results, especially their prediction about excitatory connectivity, depend on their chosen inhibitory connections? Is their choice supported by some experimental evidence? Did the authors explore inhibitory connections with a range longer than the source column but still shorter than the excitatory range?

Would the results differ qualitatively if some structure in the connectivity was included, for instance due to orientation selective columns? I think the paper would be strengthen by a discussion on this topic.

Minor suggestions and typos:

1) Page 20. Left column, first paragraph.“Only the fast receptors, AMPA and GABA, have timescales relevant to gamma band oscillations. These receptors have very fast rise times, which correspond to frequencies much higher than the gamma band. We therefore ignored the rise times of all receptors.“

This sentence sounds a bit unclear. The authors may want to mention, as they do in the main text, that they also ignored the rise times of NMDA currents since they are significantly faster than the characteristic timescales of gamma oscillations.

2) Page 20, right column. In the first paragraph, the authors clearly justify the reasons why a static input-output transfer function is a good approximation, given the synaptic dynamics they consider. However, at the end of the following paragraph, they add:

“However, as long as those gain filters are feature-less over the gamma band, their frequency dependence would not qualitatively affect the location of the gamma peak and its stimulus dependence. Thus we expect that the static I/O approximation will not alter our qualitative results.”

The authors might provide clearer insights into what they mean by 'feature-less over the gamma band'. Is this in reference to the weak dependence of responses on frequency due to the amplitude of synaptic noise and its decay time?"

3) Fig. 2. Caption: The statement 'Panels A-E and F-J show results for the columnar and non-columnar models, respectively' seems to be misplaced. Figure 2 only showcases panels A-E, which are related to the non-retinotopic model.

4) Fig. 5. Caption. Last sentence. Typo: ….H and;

5) Methods, two lines above eq. 13. Typo: “N -dimensional vector of inputs vvt”

6) Methods, two lines below eq. 13. Typo: “and f acts element-wise”. Is it f or F?

Reviewer #2: This paper investigates a model of V1 with supralinear gain function (same for all E, I populations), featuring both intra-column and inter-column horizontal connections, which captured the contrast dependence of gamma peak frequency, surround suppression and the local contrast dependence of gamma oscillations. They consider a retinotopic E-I network, with each pair of E-I populations representing a hypercolumn in the visual cortex. Thus, the eigenmode of one column could affect the spectrum of other columns depending on the inter-column connectivity, which has been studied explicitly in the method section. In addition to the receptive field distance-dependent inter-column connectivity, a strong intra-column connection is added, which they show critical to capture local contrast dependence, such that it is the local stimulus contrast that determines the peak frequency of gamma oscillations at a cortical location under a stimulus with non-uniform contrast. The paper is written clearly and is easy to read. The figures are presented clearly and analytical work is laid out reasonably well. The effort of modeling retinotopic cortical space is a necessary extension to existing models in order to both explain existing data and make any novel predictions to test the model.

This reviewer is however not convinced that there are clear predictions from the study that help us validate or reject the underlying mechanism of increased gamma power observed in LFPs. The study considers an E-I network which even under strong external input caused by increasing contrast, reaches a stable state in the absence of noise while exhibits transient (damped) oscillation under noise. This requires the system to be close to, but below a Hopf bifurcation without noise. The authors argue that 'Gamma oscillations do not behave like sustained oscillations,… as they are not auto-coherent and their timing and duration vary stochastically, resulting in a single broad peak in the power-spectrum, with no visible higher harmonics, consistent with transient (damped) and noise driven oscillations'. However, there is a body of literature that models gamma oscillation as a noisy ISN limit cycles in stochastic models (Benayoun et al. 2010, Wallace et al. 2011, Dumont et al, 2016, Li et al, 2022, etc).. These models involve a Hopf bifurcation and capture many statistical properties of gamma oscillations. It is possible to see broad peaks in model networks with noisy limit cycles in an ISN/SSN (see above), which can vary in strength and central frequency as a function of not only network parameters, but input to the network: the imaginary part of eigenvalues vary with increasing external input, similar to this proposal. These properties appear to capture many aspects of visually evoked gamma when the neural transfer function is operating in a region of accelerating nonlinearity (Veit et al, Jadi & Sejnowski). In the current study, the authors analyze the power-spectrum by applying a linearization scheme: they first find the stable point under a noise-free system and then perturb it with noise and noise-drive deviations. Then by analyzing the corresponding Fourier spectrum with Green function, they calculate the contribution of each individual eigenmode to the power spectrum characterized by ratio of LFP to the noise power spectrum, demonstrating that modes with eigenvalues having a less negative real part make stronger contribution. Whereas in the frame of noisy limit cycles in an SSN/ISN, the more positive the real part of the eigenvalue is, the greater the amplitude. It is not clear if a systematic analysis for detection of higher harmonics has been conducted on electrophysiological data (visible harmonics?). Thus it is not clear to the reviewer what makes the underlying mechanism a better choice. This is a reasonable alternate model for the observed peaks in power spectrum of LFP in visual cortex. What the reviewer would like to see additionally is a proposal for one or more experiments to test model predictions that can help support or reject this or the above mentioned alternate underlying mechanisms in a concrete way. Alternately, the authors could discuss how a Hopf bifurcation based mechanism in this retinotopic model would or would not recapitulate the experimental findings. What would possibly need to change in terms of parameters or connectivity? It is possible that either mechanisms of broad peaks (damped oscillation/noisy limit cycles in superlinear ISNs) in this retinotopic model would work. It is possible that depending on stimulus properties (large vs small, spatial frequency, etc) the underlying mechanism could switch (network on either side of Hopf bifurcation). If there are distinct model predictions for say what would happen when you change contrast of small vs large stimuli, it is very testable in current experiments. This is important because we still don’t quite understand a functional role for gamma dynamics that give these broad peaks in a clear way. They could very well be diagnostic of underlying operating regime (Ray & Maunsell, 2010). The modeling will then be valuable (as something not possible experimentally) and shed light on this issue. Future studies could then explore other non-oscillatory implications of these visual cortical networks going in and out of regimes under different patterns of stimulation.

There is at least one typo in the method section (P20, left column 16th line: change 'w' to 'w_\\alpha').

**Have the authors made all data and (if applicable) computational code underlying the findings in their manuscript fully available?**

Reviewer #1: **No: **The code is not available yet, but the authors say that a cloud repository containing all used code will be made fully available before publication and referenced in the manuscript.

Reviewer #2: Yes

PLOS authors have the option to publish the peer review history of their article (what does this mean?). If published, this will include your full peer review and any attached files.

Reviewer #1: No

Reviewer #2: No
---

## [Decision Letter · Decision Letter 1]

23 May 2024

Dear Dr. Ahmadian,

We are pleased to inform you that your manuscript 'The stabilized supralinear network accounts for the contrast dependence of visual cortical gamma oscillations' has been provisionally accepted for publication in PLOS Computational Biology.

Best regards,

Tatiana Engel

Guest Editor

PLOS Computational Biology

Thomas Serre

Section Editor

PLOS Computational Biology

Reviewer's Responses to Questions

**Comments to the Authors:**

Reviewer #1: My questions raised in the previous review have been satisfactorily addressed. This version of the manuscript incorporating new analysis on the network dynamics above the Hopf bifurcation represents a significant improvement over the original submission.

Reviewer #2: The authors have satisfactorily addressed the concerns raised in the prior review. No further comments on the revision.

**Have the authors made all data and (if applicable) computational code underlying the findings in their manuscript fully available?**

Reviewer #1: Yes

Reviewer #2: None

PLOS authors have the option to publish the peer review history of their article (what does this mean?). If published, this will include your full peer review and any attached files.

Reviewer #1: No

Reviewer #2: No

---

## [Editor Report · Acceptance letter]

17 Jun 2024

PCOMPBIOL-D-23-01209R1 

The stabilized supralinear network accounts for the contrast dependence of visual cortical gamma oscillations

Dear Dr Ahmadian,

I am pleased to inform you that your manuscript has been formally accepted for publication in PLOS Computational Biology. Your manuscript is now with our production department and you will be notified of the publication date in due course.

With kind regards,

Zsofia Freund
